# Faster Perturbed Stochastic Gradient Methods for Finding Local Minima

## Abstract

Escaping from saddle points and finding local minimum is a central problem in nonconvex optimization. Perturbed gradient methods are perhaps the simplest approach for this problem. However, to find $(\epsilon, \sqrt{\epsilon})$-approximate local minima, the existing best stochastic gradient complexity for this type of algorithms is $\widetilde{O}(\epsilon^{-3.5})$, which is not optimal. In this paper, we propose `Pullback`, a faster perturbed stochastic gradient framework for finding local minima. We show that Pullback with stochastic gradient estimators such as SARAH/SPIDER and STORM can find $(\epsilon, \epsilon_H)$-approximate local minima within $\widetilde{O}(\epsilon^{-3} + \epsilon_H^{-6})$ stochastic gradient evaluations (or $\widetilde{O}(\epsilon^{-3})$ when $\epsilon_H = \sqrt{\epsilon}$). The core idea of our framework is a step-size "pullback" scheme to control the average movement of the iterates, which leads to faster convergence to the local minima.

## 1 Introduction

In this paper, we focus on the following optimization problem

$$\min_{\mathbf{x} \in \mathbb{R}^d} F(\mathbf{x}) := \mathbb{E}_{\boldsymbol{\xi}}[f(\mathbf{x}; \boldsymbol{\xi})], \tag{1.1}$$

where $f(\mathbf{x}; \boldsymbol{\xi}) : \mathbb{R}^d \to \mathbb{R}$ is a stochastic function indexed by some random vector $\boldsymbol{\xi}$, and it is differentiable and possibly nonconvex. We consider the case where only the stochastic gradients $\nabla f(\mathbf{x}; \boldsymbol{\xi})$ are accessible. (1.1) can unify a variety of stochastic optimization problems, such as finite-sum optimization and online optimization. Since in general, finding global minima of a nonconvex function could be an NP-hard problem [12], one often seeks to finding an $(\epsilon, \epsilon_H)$-approximate local minimum $\mathbf{x}$, i.e., $\|\nabla F(\mathbf{x})\|_2 \leq \epsilon$ and $\lambda_{\min}\big(\nabla^2 F(\mathbf{x})\big) \geq -\epsilon_H$, where $\nabla F(\mathbf{x})$ is the gradient of $F$ and $\lambda_{\min}\big(\nabla^2 F(\mathbf{x})\big)$ is the smallest eigenvalue of the Hessian of $F$ at $\mathbf{x}$. In many machine learning applications such as matrix sensing and completion [5, 11], it suffices to find local minima due to the fact that all local minima are global minima.

For the case where $f$ is a deterministic function, it has been shown that vanilla gradient descent fails to find local minima efficiently since the iterates will get stuck at saddle points for exponential time [8]. To address this issue, the simplest idea is to add random noises as a perturbation to the stuck iterates. Jin et al. [13] showed that the simple perturbation step is enough for gradient descent to escape saddle points and find $(\epsilon, \sqrt{\epsilon})$-approximate local minima within $\widetilde{O}(1/\epsilon^2)$ gradient evaluations, which matches the number of gradient evaluations for gradient descent to find $\epsilon$-stationary points [19]. Such matching results suggest that perturbed gradient methods can find local minima efficiently, at least for deterministic optimization. When it comes to stochastic optimization, a natural question arises:

*Can perturbed **stochastic** gradient methods find local minima efficiently?*

To answer this question, we first look into existing results of perturbed stochastic gradient methods for finding local minima. Ge et al. [10] showed that perturbed Stochastic gradient descent can find

35  $(\epsilon, \sqrt{\epsilon})$-approximate local minima within $\widetilde{O}(\text{poly}(\epsilon^{-1}))$ stochastic gradient evaluations. Daneshmand
36  et al. [7] showed that under a specific CNC condition, stochastic gradient descent is able to find
37  $(\epsilon, \sqrt{\epsilon})$-approximate local minima within $\widetilde{O}(1/\epsilon^5)$ stochastic gradient evaluations. Later on, Li [17]
38  showed that simple stochastic recursive gradient descent (SSRGD) can find $(\epsilon, \sqrt{\epsilon})$-approximate
39  local minima within $\widetilde{O}(\epsilon^{-3.5})$ stochastic gradient evaluations, which is the state-of-the-arts to date.
40  However, none of these results by perturbed stochastic gradient methods matches the optimal result
41  $\widetilde{O}(\epsilon^{-3})$ for finding $\epsilon$-stationary points, achieved by SPIDER [9], SNVRG [30] and STORM [6] (See
42  also Arjevani et al. [4] for the lower bound results). Therefore, whether perturbed stochastic gradient
43  methods can find local minima as efficiently as finding stationary points still remains unknown.

44  In this work, we give an affirmative answer to the above question. We propose a general framework
45  named `Pullback`, which works together with existing popular stochastic gradient estimators such
46  as SARAH/SPIDER and STORM to find approximate local minima efficiently. We summarize our
47  contributions as follows:

48  • We prove that `Pullback` finds $(\epsilon, \epsilon_H)$-approximate local minima within $\widetilde{O}(\epsilon^{-3} + \epsilon_H^{-6})$ stochastic
49    gradient evaluations. Specifically, in the classic setting where $\epsilon_H = \sqrt{\epsilon}$, our `Pullback` together
50    with the SARAH/SPIDER estimator enjoys an $\widetilde{O}(\epsilon^{-3})$ stochastic gradient complexity, which
51    outperforms previous best known complexity result $\widetilde{O}(\epsilon^{-3.5})$ achieved by Li [17]. Our result
52    also matches the best possible complexity result $\widetilde{O}(\epsilon^{-3})$ achieved by negative curvature search
53    based algorithms [9, 31], which suggests that *simple* methods such as perturbed stochastic gradient
54    methods can find local minima as efficiently as the more complicated ones.

55  • Besides, we also show that `Pullback` with a recent proposed STORM estimator is also able to find
56    $(\epsilon, \epsilon_H)$-approximate local minima within $\widetilde{O}(\epsilon^{-3} + \epsilon_H^{-6})$ stochastic gradient evaluations.

57  • At the core of our `Pullback` is a novel step-size "pullback" scheme to control the average movement
58    of the iterates, which may be of independent interest to other related nonconvex optimization
59    algorithm design.

60  To compare with, we summarized related results of stochastic first-order methods for finding local
61  minima in Table 1.

Table 1: Comparison of of different optimization algorithm for find approximate local minima of non convex online problems.

| Algorithm | Gradient complexity | Classic Setting | Neon2 |
|---|---|---|---|
| Neon2+Natasha2 [1] | $\widetilde{\mathcal{O}}(\epsilon^{-3.25} + \epsilon^{-3}\epsilon_H^{-1} + \epsilon_H^{-5})$ | $\widetilde{\mathcal{O}}(\epsilon^{-3.5})$ | needed |
| Neon2+SCSG [3] | $\widetilde{\mathcal{O}}(\epsilon^{-10/3} + \epsilon^{-2}\epsilon_H^{-3} + \epsilon_H^{-5})$ | $\widetilde{\mathcal{O}}(\epsilon^{-3.5})$ | needed |
| SNVRG$^+$+Neon2 [31] | $\widetilde{\mathcal{O}}(\epsilon^{-3} + \epsilon^{-2}\epsilon_H^{-3} + \epsilon_H^{-5})$ | $\widetilde{\mathcal{O}}(\epsilon^{-3.5})$ | needed |
| SPIDER-SFO$^+$(+Neon2)[9] | $\widetilde{\mathcal{O}}(\epsilon^{-3} + \epsilon^{-2}\epsilon_H^{-2} + \epsilon_H^{-5})$ | $\widetilde{\mathcal{O}}(\epsilon^{-3})$ | needed |
| Perturbed SGD [10] | $\text{Poly}(d, \epsilon^{-1}, \epsilon_H^{-1})$ | $\widetilde{\mathcal{O}}(\text{Poly}(\epsilon^{-1}))$ | No |
| CNC-SGD [7] | $\widetilde{\mathcal{O}}(\epsilon^{-4} + \epsilon_H^{-10})$ | $\widetilde{\mathcal{O}}(\epsilon^{-5})$ | No |
| SSRGD [17] | $\widetilde{\mathcal{O}}(\epsilon^{-3} + \epsilon^{-2}\epsilon_H^{-3} + \epsilon^{-1}\epsilon_H^{-4})$ | $\widetilde{\mathcal{O}}(\epsilon^{-3.5})$ | No |
| Pullback (**This paper**) | $\widetilde{\mathcal{O}}(\epsilon^{-3} + \epsilon_H^{-6})$ | $\widetilde{\mathcal{O}}(\epsilon^{-3})$ | No |

62  **Notations** We use lower case letters to denote scalars, lower and upper case bold letters to denote
63  vectors and matrices. We use $\|\cdot\|$ to indicate Euclidean norm. We use $\mathbb{B}_{\mathbf{x}}(r)$ to denote a Euclidean
64  ball center at $\mathbf{x}$ with radius $r$. We also use the standard $O$ and $\Omega$ notations. We use $\lambda_{\min}(\mathbf{M})$ to denote
65  the minimum eigenvalue of matrix $\mathbf{M}$. We say $a_n = O(b_n)$ if and only if $\exists C > 0, N > 0, \forall n >$
66  $N, a_n \leq Cb_n$; $a_n = \Omega(b_n)$ if $a_n \geq Cb_n$. The notation $\widetilde{O}$ is used to hide logarithmic factors.

## 2 Related Work

68  In this section, we review some important related works.

69  **Variance reduction methods for finding stationary points.** Our algorithm takes stochastic gradient
70  estimators as its subroutine. In specific, Johnson and Zhang [14], Xiao and Zhang [28] proposed
71  Stochastic Variance Reduced Gradient (SVRG) for convex optimization in the finite-sum setting.
72  Reddi et al. [25], Allen-Zhu and Hazan [2] analyzed SVRG for nonconvex optimization. Lei et al. [16]

proposed a new variance reduction algorithm, dubbed stochastically controlled stochastic gradient (SCSG) algorithm, which finds a $\epsilon$-stationary point within $O(\epsilon^{-10/3})$ stochastic gradient evaluations. Nguyen et al. [21] proposed a SARAH algorithm which uses a recursive gradient estimator for convex optimization, and it was extended to nonconvex optimization in [22]. Fang et al. [9] proposed a SPIDER algorithm with a recursive gradient estimator and proved an $O(\epsilon^{-3})$ stochastic gradient evaluations to find a $\epsilon$-stationary point, which matches a corresponding lower bound. Concurrently, Zhou et al. [30] proposed an SNVRG algorithm with a nested gradient estimator and proved an $\widetilde{O}(\epsilon^{-3})$ stochastic gradient evaluations to find a $\epsilon$-stationary point. Wang et al. [27] proposed a Spiderboost algorithm with a constant step size, achieves the same $O(\epsilon^{-3})$ gradient complexity. Pham et al. [23] extended SARAH [22] to proximal optimization and proved $O(\epsilon^{-3})$ gradient complexity for finding stationary points. Recently, Cutkosky and Orabona [6] proposed a recursive momentum-based algorithm called STORM and proved an $\widetilde{O}(\epsilon^{-3})$ gradient complexity to find $\epsilon$-stationary points. Tran-Dinh et al. [26] proposed a SARAH-SGD algorithm which hybrids both SGD and SARAH algorithm with an $\widetilde{O}(\epsilon^{-3})$ gradient complexity when $\epsilon$ is small. Li et al. [18] proposed a PAGE algorithm with probabilistic gradient estimator which also attains an $\widetilde{O}(\epsilon^{-3})$ gradient complexity. In our work, we employ SARAH/SPIDER and STORM as the gradient estimator in our `Pullback` framework since they are most representative and simple to use.

**Utilizing negative curvature descent to escape from saddle points.** To escape saddle points, a widely used approach is to first compute the direction of the negative curvature of the saddle point and move away along that direction. In stochastic optimization, to find $(\epsilon, \epsilon_H)$-approximate local minima, [1] proposed a Natasha algorithm using Hessian-vector product to compute the negative curvature direction with the total computation cost of $\widetilde{O}(\epsilon^{-3.25} + \epsilon^{-3}\epsilon_H^{-1} + \epsilon_H^{-5})$. Later, Xu et al. [29] proposed a Neon method which computes the negative curvature direction with perturbed stochastic gradients, whose total computational cost is $\widetilde{O}(\epsilon^{-10/3} + \epsilon^{-2}\epsilon_H^{-3} + \epsilon_H^{-6})$. [3] proposed a Neon2 negative curvature computation subroutine with $\widetilde{O}(\epsilon^{-10/3} + \epsilon^{-2}\epsilon_H^{-3} + \epsilon_H^{-5})$ stochastic gradient evaluations. Fang et al. [9] then showed that SPIDER equipped with Neon2 can find $(\epsilon, \epsilon_H)$-approximate local minima within $\widetilde{\mathcal{O}}(\epsilon^{-3} + \epsilon^{-2}\epsilon_H^{-2} + \epsilon_H^{-5})$ stochastic gradient evaluations, while independently Zhou et al. [31] proved that SNVRG equipped with Neon2 can find $(\epsilon, \epsilon_H)$-approximate local minima within $\widetilde{\mathcal{O}}(\epsilon^{-3} + \epsilon^{-2}\epsilon_H^{-3} + \epsilon_H^{-5})$ stochastic gradient evaluations. In contrast to this line of works, our algorithm is simpler since it does not need to use the negative curvature search routine.

# 3 Preliminaries

In this section, we present assumptions and definitions that will be used throughout our analysis.

We first introduce the standard smoothness and Hessian Lipschitz assumptions.

**Assumption 3.1.** For all $\boldsymbol{\xi}$, $f(\cdot; \boldsymbol{\xi})$ is $L$-smooth and its Hessian is $\rho$-Lipschitz continuous w.r.t. $\mathbf{x}$, i.e., for any $\mathbf{x}_1, \mathbf{x}_2$, we have that

$$\|\nabla f(\mathbf{x}_1; \boldsymbol{\xi}) - \nabla f(\mathbf{x}_2; \boldsymbol{\xi})\|_2 \le L\|\mathbf{x}_1 - \mathbf{x}_2\|_2, \ \|\nabla^2 f(\mathbf{x}_1; \boldsymbol{\xi}) - \nabla^2 f(\mathbf{x}_2; \boldsymbol{\xi})\|_2 \le \rho\|\mathbf{x}_1 - \mathbf{x}_2\|_2$$

This assumption directly implies that the expected objective function $F(\mathbf{x})$ is also $L$-smooth and its Hessian is $\rho$-Lipschitz continuous. This assumption is standard for finding approximate local minima in all the results presented in Table 1.

**Assumption 3.2.** The squared difference between the stochastic gradient and full gradient is bounded by $\sigma^2 < \infty$, i.e., for any $\mathbf{x}, \boldsymbol{\xi} \in \mathbb{R}^d$, $\|\nabla f(\mathbf{x}; \boldsymbol{\xi}) - \nabla F(\mathbf{x})\|_2^2 \le \sigma^2$.

Assumption 3.2 is standard in online/stochastic optimization for finding second-order stationary points [9, 17], and immediately implies that the variance of the stochastic gradient is bounded by $\sigma^2$. It can be relaxed to be $\|\nabla f(\mathbf{x}; \boldsymbol{\xi}) - \nabla F(\mathbf{x})\|_2$ has a $\sigma$-Sub-Gaussian tail.

Let $\mathbf{x}_0 \in \mathbb{R}^d$ be the starting point of the algorithm. We assume the gap between the initial function value and the optimal value is bounded.

**Assumption 3.3.** We have $\Delta = F(\mathbf{x}_0) - \inf_{\mathbf{x}} F(\mathbf{x}) < +\infty$.

Next, we give the formal definition of approximate local minima (a.k.a., second-order stationary points).

**Definition 3.4.** We call $\mathbf{x} \in \mathbb{R}^d$ an $(\epsilon, \epsilon_H)$-approximate local minimum, if

$$\|\nabla F(\mathbf{x})\|_2 \leq \epsilon, \lambda_{\min}\big(\nabla^2 F(\mathbf{x})\big) \geq -\epsilon_H.$$

The definition of $(\epsilon, \epsilon_H)$-approximate local minima is a generalization of the classical $(\epsilon, \sqrt{\epsilon})$-approximate local minima studied by Nesterov and Polyak [20], Jin et al. [13].

## 4  The `Pullback` Framework

In this section, we present our main algorithm `Pullback`. We begin with reviewing the mechanism of perturbed gradient descent in deterministic optimization, and then we discuss the main difficulty of extending it to the stochastic optimization case. Finally, we show how we overcome such a difficulty by presenting our `Pullback` framework.

**How does perturbed gradient descent escape from saddle points?**  We review the perturbed gradient descent [13] (PGD for short) with its proof roadmap, which shows how PGD finds $(\epsilon, \sqrt{\epsilon})$-approximate local minima efficiently. In general, the whole process of perturbed gradient descent can be decomposed into several epochs, and each epoch consists of two non-overlapping phases: the *gradient descent phase* (GD phase for short) and the *Escape from saddle point phase* (Escape phase for short). In each epoch, PGD starts with the GD phase by default. In the GD phase, PGD performs vanilla gradient descent to update its iterate, until at some iterate $\widetilde{\mathbf{x}}$, the norm of the gradient $\|\nabla F(\widetilde{\mathbf{x}})\|_2$ is less than the target accuracy $\widetilde{O}(\epsilon)$. Then PGD switches to the Escape phase. In the Escape phase, PGD first adds a uniform random noise (or Gaussian noise) to the current iterate $\widetilde{\mathbf{x}}$, then it runs $\ell_{\mathrm{thres}} = \widetilde{O}(\epsilon^{-1/2})$ steps of vanilla gradient descent. PGD then compares the function value gap between the current iterate and the beginning iterate of Escape phase $\widetilde{\mathbf{x}}$. If the gap is less than a threshold $\mathcal{F} = \widetilde{O}(\epsilon^{1.5})$, then PGD outputs $\widetilde{\mathbf{x}}$ as the targeted local minimum. Otherwise, PGD starts a new epoch and performs gradient descent again.

To see why PGD can find $(\epsilon, \sqrt{\epsilon})$-approximate local minima within $\widetilde{O}(\epsilon^{-2})$ gradient evaluations, we do the following calculation. First, when PGD is in the GD phase, the function value decreases $\widetilde{O}(\epsilon^2)$ per step (following the standard gradient descent analysis). When PGD is in the Escape phase, the function value decreases $\mathcal{F}/\ell_{\mathrm{thres}} = \widetilde{O}(\epsilon^2)$ per step on average. Therefore, the total number of steps will be bounded by $\widetilde{O}(\epsilon^{-2})$, which is in the same order as GD for finding $\epsilon$-stationary points.

**Limitation of existing methods.** However, extending the two-phase PGD algorithm from deterministic optimization to stochastic optimization with a competative gradient complexity is very challenging. We take the SSRGD algorithm proposed by Li [17] as an example, which uses SARAH/SPIDER [9] as its gradient estimator. Unlike deterministic optimization where we can access the exact function value $F(\mathbf{x})$ and gradient $\nabla F(\mathbf{x})$ defined in (1.1), in the stochastic optimization case we can only access the stochastic function $f(\mathbf{x}; \boldsymbol{\xi})$ and the stochastic gradient $\nabla f(\mathbf{x}; \boldsymbol{\xi})$. Therefore, in order to estimate the gradient norm $\|\nabla F(\mathbf{x})\|_2$ (which is required at the beginning of Escape phase), a naive approach (adapted by Li [17]) is to sample a big batch of stochastic gradient $\nabla f(\mathbf{x}; \boldsymbol{\xi}_1), \ldots, \nabla f(\mathbf{x}; \boldsymbol{\xi}_B)$ and uses their mean to approximate $\nabla F(\mathbf{x})$. Standard concentration analysis suggests that in order to achieve an $\epsilon$-accuracy, the batch size $B$ should be in the order $\widetilde{O}(\epsilon^{-2})$. Thus, each Escape phase leads to a $\mathcal{F} = \widetilde{O}(\epsilon^{1.5})$ function value decrease with at least $\widetilde{O}(\epsilon^{-2})$ number of stochastic gradient evaluations, which contributes $\widetilde{O}(1/\epsilon^{1.5} \cdot \epsilon^{-2}) = \widetilde{O}(\epsilon^{-3.5})$ gradient complexity in the end. This is already worse than the $O(\epsilon^{-3})$ gradient complexity of SPIDER for finding $\epsilon$-stationary points.

**Our approach.** Here we propose our `Pullback` framework in Algorithm 1, which overcomes the aforementioned limitation. In detail, `Pullback` inherits the two-phase structure of PGD and SSRGD, and it takes either SARAH/SPIDER or STORM [6] as its gradient estimator. The two gradient estimators are summarized as subroutines `GradEst-SPIDER` and `GradEst-STORM` in Algorithms 2 and 3 respectively, and we use $\mathbf{d}_t$ to denote their estimated gradient at iterate $\mathbf{x}_t$. The key improvement of `Pullback` is that, it directly takes the output of the gradient estimator `GradEst` to estimate the true gradient $\nabla F(\mathbf{x})$, which avoids sampling a big batch of stochastic gradients as in Li [17] and thus saves the total gradient complexity. A similar strategy has also been adapted in [9], but for the negative curvature search subroutine. However, such a strategy leads to a new problem to be solved.

Since we use $\mathbf{d}_t$ to directly estimate $\nabla F(\mathbf{x}_t)$, in order to make such an estimation valid, we need to guarantee that the error between $\mathbf{d}_t$ and $\nabla F(\mathbf{x}_t)$ is small enough, e.g., up to $O(\epsilon)$ accuracy. Notice

---

**Algorithm 1** `Pullback`

---

**Input:** Initial point $\mathbf{x}_1$, step size $\eta$ and $\eta_H$, perturbation radius $r$, threshold parameter $\ell_{\text{thres}}$, average movement $\overline{D}$.

1: $\mathbf{d}_1 \leftarrow \texttt{GradEst}(0, \mathbf{0}, \mathbf{0}, \mathbf{x}_1)$, $s \leftarrow 0$, $t \leftarrow 1$, FIND$\leftarrow$false
2: **while** FIND = false **do**
3: $\quad$ $s \leftarrow s + 1$, $t_s \leftarrow t$, FIND$\leftarrow$true
4: $\quad$ **while** $\|\mathbf{d}_t\|_2 > \epsilon$ **do**
5: $\quad\quad$ $\eta_t \leftarrow \eta/\|\mathbf{d}_t\|_2$, {"PullBack"}
6: $\quad\quad$ $\mathbf{x}_{t+1} \leftarrow \mathbf{x}_t - \eta_t\mathbf{d}_t$, $\mathbf{d}_{t+1} \leftarrow \texttt{GradEst}(t, \mathbf{d}_t, \mathbf{x}_t, \mathbf{x}_{t+1})$, $t \leftarrow t + 1$
7: $\quad$ **end while**
8: $\quad$ $m_s \leftarrow t$, $\boldsymbol{\xi} \sim \text{Uniform } B_0(r)$, $\mathbf{x}_{t+1} \leftarrow \mathbf{x}_t + \boldsymbol{\xi}$, $\mathbf{d}_{t+1} \leftarrow \texttt{GradEst}(t, \mathbf{d}_t, \mathbf{x}_t, \mathbf{x}_{t+1})$, $t \leftarrow t+1$
9: $\quad$ **for** $k = 0, \ldots, \ell_{\text{thres}} - 1$ **do**
10: $\quad\quad$ $\eta_t \leftarrow \eta_H$, $D \leftarrow \sum_{i=m_s}^{t} \eta_i^2\|\mathbf{d}_i\|_2^2$
11: $\quad\quad$ **if** $D > (t - m_s + 1)\overline{D}$ **then**
12: $\quad\quad\quad$ Set $\eta_t$ such that $\sum_{i=m_s}^{t} \eta_i^2\|\mathbf{d}_i\|_2^2 = (t - m_s + 1)\overline{D}$ {"PullBack"}
13: $\quad\quad\quad$ $\mathbf{x}_{t+1} \leftarrow \mathbf{x}_t - \eta_t\mathbf{d}_t$, $\mathbf{d}_{t+1} \leftarrow \texttt{GradEst}(t, \mathbf{d}_t, \mathbf{x}_t, \mathbf{x}_{t+1})$, $t \leftarrow t + 1$, FIND $\leftarrow$ false, **break**
14: $\quad\quad$ **end if**
15: $\quad\quad$ $\mathbf{x}_{t+1} \leftarrow \mathbf{x}_t - \eta_t\mathbf{d}_t$, $\mathbf{d}_{t+1} \leftarrow \texttt{GradEst}(t, \mathbf{d}_t, \mathbf{x}_t, \mathbf{x}_{t+1})$, $t \leftarrow t + 1$
16: $\quad$ **end for**
17: **end while**

**Output:** $\mathbf{x}_{m_s}$

---

**Algorithm 2** `GradEst-SPIDER`$(t, \mathbf{d}_t, \mathbf{x}_t, \mathbf{x}_{t+1}, b, q, B)$

---

**Input:** Big batch size $B$, mini-batch size $b$, loop length $q$

1: **if** $t \bmod q = 0$ **then**
2: $\quad$ Generate $\boldsymbol{\xi}_{t+1}^1, \ldots, \boldsymbol{\xi}_{t+1}^B$. Set $\mathbf{d}_{t+1} \leftarrow \sum_{i=1}^{B} \nabla f(\mathbf{x}_{t+1}; \boldsymbol{\xi}_{t+1}^i)/B$
3: **else**
4: $\quad$ Generate $\boldsymbol{\xi}_{t+1}^1, \ldots, \boldsymbol{\xi}_{t+1}^b$. Set $\mathbf{d}_{t+1} \leftarrow \mathbf{d}_t + \sum_{i=1}^{b} \left[\nabla f(\mathbf{x}_{t+1}; \boldsymbol{\xi}_{t+1}^i) - \nabla f(\mathbf{x}_t; \boldsymbol{\xi}_{t+1}^i)\right]/b$
5: **end if**

**Output:** $\mathbf{d}_{t+1}$

---

that the recursive structure of SARAH/SPIDER and STORM suggests the following error bound:

$$\forall t, \ \|\mathbf{d}_t - \nabla F(\mathbf{x}_t)\|_2^2 = \widetilde{O}\left(\sum_{i=s_t}^{t-1} \|\mathbf{x}_{i+1} - \mathbf{x}_i\|_2^2\right), \tag{4.1}$$

where $s_t$ is some reference index only related to $t$. Therefore, in order to make the error $\|\mathbf{d}_t - \nabla F(\mathbf{x}_t)\|_2$ small, it suffices to make the movement of the iterates $\|\mathbf{x}_{i+1} - \mathbf{x}_i\|_2$ small either individually or on average. We achieve this goal by our proposed step-size "Pullback" scheme. In detail, in the GD phase, when the norm of the estimated gradient $\|\mathbf{d}_t\|_2$ is large, we pull the step-size $\eta_t$ back to a smaller value via normalization, which forces the movement $\|\mathbf{x}_{i+1} - \mathbf{x}_i\|_2 = \eta_t\|\mathbf{d}_t\|_2 = \eta$ to be small. Such an approach is also adapted by Fang et al. [9] as a normalized gradient descent for finding first-order stationary points. In the Escape phase, which starts at $m_s$-th step, we record the accumulative squared movement starting from $\mathbf{x}_{m_s+1}$ (after the perturbation step) as $D := \sum_{i=m_s+1}^{t} \|\mathbf{x}_{i+1} - \mathbf{x}_i\|_2^2 = \sum_{i=m_s+1}^{t} \eta_i^2\|\mathbf{d}_i\|_2^2$. When the average movement $D/(t - m_s + 1)$ is large, we pull the *last* step size $\eta_t$ back to a smaller value, which forces the average movement $D/(t - m_s + 1)$ to be small. Fortunately, such a simple step-size calibration scheme allows us to well-control the error between $\mathbf{d}_t$ and $\nabla F(\mathbf{x}_t)$, and to reduce the gradient complexity.

## 5 Main Results

In this section, we present the main theoretical results. We first present the convergence guarantee of `Pullback-SPIDER`, which uses `GradEst-SPIDER` to estimate the gradient $\mathbf{d}_t$ in Algorithm 1.

**Theorem 5.1.** Under Assumptions 3.1, 3.2 and 3.3, choose batch size $B = \widetilde{O}(\sigma^2\epsilon^{-2} + \sigma^2\rho^2\epsilon_H^{-4})$, $b = q = \sqrt{B}$, set step size $\eta = \sigma/(2\sqrt{B}L)$, $\eta_H = \widetilde{O}(L^{-1})$, perturbation radius $r \leq \min\left\{\sigma/(2\sqrt{B}L), \log(4/\delta)\eta_H\sigma^2/(2B\epsilon), \sqrt{2\log(4/\delta)\eta_H\sigma^2/(BL)}\right\}$, threshold $\ell_{\text{thres}} = $

---

**Algorithm 3** GradEst-STORM$(t, \mathbf{d}_t, \mathbf{x}_t, \mathbf{x}_{t+1}, a, b, B)$

**Input:** Initial batch size $B$, mini batch size $b$ and weight parameter $a$.
  1: **if** $t = 0$ **then**
  2:   Generate $\boldsymbol{\xi}_{t+1}^1, \ldots, \boldsymbol{\xi}_{t+1}^B$. Set $\mathbf{d}_{t+1} \leftarrow \sum_{i=1}^{B} \nabla f(\mathbf{x}_{t+1}; \boldsymbol{\xi}_{t+1}^i)/B$
  3: **else**
  4:   Generate $\boldsymbol{\xi}_{t+1}^1, \ldots, \boldsymbol{\xi}_{t+1}^b$
  5:   Set $\mathbf{d}_{t+1} \leftarrow (1-a)\big[\mathbf{d}_t - \sum_{i=1}^{b} \nabla f(\mathbf{x}_t; \boldsymbol{\xi}_{t+1}^i)/b\big] + \sum_{i=1}^{b} \nabla f(\mathbf{x}_{t+1}; \boldsymbol{\xi}_{t+1}^i)/b$
  6: **end if**
**Output:** $\mathbf{d}_{t+1}$

---

$\widetilde{O}(1/(\eta_H \epsilon_H))$ and $\overline{D} = \sigma^2/(4BL^2)$. Then with high probability, Pullback-SPIDER can find $(\epsilon, \epsilon_H)$-approximate local minima within $\widetilde{O}\big(\sigma L \Delta \epsilon^{-3} + \sigma \rho^3 L \Delta \epsilon_H^{-6}\big)$ stochastic gradient evaluations.

**Remark 5.2.** In the classical setting $\epsilon = \sqrt{\epsilon_H}$, our result gives $\widetilde{O}(\epsilon^{-3})$ gradient complexity, which outperforms the best existing result $\widetilde{O}(\epsilon^{-3.5})$ for perturbed stochastic gradient methods achieved by SSRGD [17]. For sufficiently small $\epsilon$, Arjevani et al. [4] proved the lower bound of gradient complexity $\Omega(\epsilon^{-3} + \epsilon_H^{-5})$ for any first-order stochastic methods to find $(\epsilon, \epsilon_H)$-approximate local minima. Our results matches the lower bound $\widetilde{O}(\epsilon^{-3})$ when $\epsilon_H \leq \epsilon^{3/5}$. For the general case, there is still a gap in the dependency of $\epsilon_H$ between our result and the lower bound, and we leave to close it as future work.

Next, we present the convergence guarantee of Pullback-STORM, which uses GradEst-STORM to estimate the gradient $\mathbf{d}_t$ in Algorithm 1.

**Theorem 5.3.** Under Assumptions 3.1, 3.2 and 3.3, choose the mini batch size $b = \widetilde{O}(\sigma \epsilon^{-1} + \sigma \rho \epsilon_H^{-2})$, and initial batch size $B = b^2$, set step size $\eta = \sigma/(2bL)$, $\eta_H = \widetilde{O}(L^{-1})$, weight $a = 56^2 \log(4/\delta)/b$, threshold $\ell_{\text{thres}} = \widetilde{O}(1/(\eta_H \epsilon_H))$, perturbation radius $r \leq \min\big\{\sigma/(2bL), \log(4/\delta)^2 \eta_H \sigma^2/(\epsilon b^2), \sqrt{2\log(4/\delta)^2 \eta_H \sigma^2/(b^2 L)}\big\}$, and $\overline{D} = \sigma^2/(4b^2 L^2)$. Then with high probability, Pullback-STORM can find $(\epsilon, \epsilon_H)$-approximate local minima within $\widetilde{O}\big(\sigma L \Delta \epsilon^{-3} + \sigma \rho^3 L \Delta \epsilon_H^{-6}\big)$ stochastic gradient evaluations.

**Remark 5.4.** Different from Pullback-SPIDER, the estimation error $\|\mathbf{d}_t - \nabla F(\mathbf{x}_t)\|_2$ of Pullback-STORM is controlled by the weight parameter $a$. This allows us to come up with a simpler single-loop algorithm instead of a double-loop algorithm.

## 6  Proof Outline of the Main Results

Due to the page limit, we only outline the proof of Theorem 5.1 and leave the proof of Theorem 5.3 to the appendix.

Let $\boldsymbol{\epsilon}_t$ denote the difference between true gradient $\nabla F(\mathbf{x}_t)$ and the estimated gradient $\mathbf{d}_t$, which is $\boldsymbol{\epsilon}_t := \mathbf{d}_t - \nabla F(\mathbf{x}_t)$. The following lemma suggests that the estimation error $\|\boldsymbol{\epsilon}_t\|_2$ can be bounded.

**Lemma 6.1.** Under Assumptions 3.1 and 3.2, set $b = q = \sqrt{B}$, $\eta \leq \sigma/(2\sqrt{B}L)$, $r \leq \sigma/(2\sqrt{B}L)$ and $\overline{D} \leq \sigma^2/(4BL^2)$, then with probability at least $1 - \delta$, for all $t$ we have

$$\|\boldsymbol{\epsilon}_t\|_2 \leq \sqrt{8\log(4/\delta)}\sigma/\sqrt{B}.$$

Specifically, by the choice of $B$ in Theorem 5.1 we have that $\|\boldsymbol{\epsilon}_t\|_2 \leq \epsilon/2$.

*Proof of Lemma 6.1.* By GradEst-SPIDER presented in Algorithm 2 we have

$$\boldsymbol{\epsilon}_{t+1} = \frac{1}{B} \sum_{i=1}^{B} \big[\nabla f(\mathbf{x}_{t+1}; \boldsymbol{\xi}_{t+1}^i) - \nabla F(\mathbf{x}_{t+1})\big], \qquad\qquad t \bmod q = 0,$$

$$\boldsymbol{\epsilon}_{t+1} = \boldsymbol{\epsilon}_t + \frac{1}{b} \sum_{i=1}^{b} \big[\nabla f(\mathbf{x}_{t+1}; \boldsymbol{\xi}_{t+1}^i) - \nabla f(\mathbf{x}_t; \boldsymbol{\xi}_{t+1}^i) - \nabla F(\mathbf{x}_{t+1}) + \nabla F(\mathbf{x}_t)\big], \quad t \bmod q \neq 0.$$

By the $L$-smoothness in Assumption 3.1 we have

$$\big\|\nabla f(\mathbf{x}_{t+1}; \boldsymbol{\xi}_{t+1}^i) - \nabla f(\mathbf{x}_t; \boldsymbol{\xi}_{t+1}^i) - \nabla F(\mathbf{x}_{t+1}) + \nabla F(\mathbf{x}_t)\big\|_2 \leq 2L\|\mathbf{x}_{t+1} - \mathbf{x}_t\|_2.$$

Then by Assumption 3.2 and Azuma–Hoeffding inequality (See Lemma D.1 for details), with probability at least $1 - \delta$, we have

$$\forall t > 0, \ \|\boldsymbol{\epsilon}_{t+1}\|_2^2 \leq 4\log(4/\delta)\left(\frac{\sigma^2}{B} + \frac{4L^2}{b}\sum_{i=\lfloor t/q \rfloor q}^{t} \|\mathbf{x}_{i+1} - \mathbf{x}_i\|_2^2\right). \tag{6.1}$$

Notice that `GradEst-SPIDER` is parallel with `Pullback`. Thus we need to further bound (6.1) by considering iterates in three different cases: (1) for step $i$ in the GD phase, we have $\|\mathbf{x}_{i+1} - \mathbf{x}_i\|_2^2 \leq \eta^2$ due to the "Pullback" scheme; (2) for $i = m_s$ for some $s$ in the Escape phase, we have $\|\mathbf{x}_{i+1} - \mathbf{x}_i\|_2^2 \leq r^2$; and (3) for the other steps in Escape phase, we have on average, $\|\mathbf{x}_{i+1} - \mathbf{x}_i\|_2^2 \leq \overline{D}$. Therefore we have

$$\|\boldsymbol{\epsilon}_{t+1}\|_2^2 \leq 4\log(4/\delta)\left(\frac{\sigma^2}{B} + \frac{4L^2}{b} \cdot q \cdot \max\{\eta^2, r^2, \overline{D}\}\right) \leq \frac{8\log(4/\delta)\sigma^2}{B}.$$

$\square$

Lemma 6.1 guarantees that with high probability $\|\nabla F(\mathbf{x}_t)\|_2 \leq \|\mathbf{d}_t\|_2 + \epsilon$, which ensures $\|\nabla F(\mathbf{x}_{m_s})\|_2 \leq 2\epsilon$ when the algorithm terminates. Next lemma bounds the function value decrease in the GD phase, which is also valid for `Pullback-STORM`.

**Lemma 6.2.** Suppose the event in Lemma 6.1 holds, $\eta \leq \epsilon/(2L)$, then for any $s$, we have

$$F(\mathbf{x}_{t_s}) - F(\mathbf{x}_{m_s}) \geq \frac{(m_s - t_s)\eta\epsilon}{8}.$$

The choice of $\eta$ in Theorem 5.1 further implies that the loss decreases by at least $\sigma\epsilon/(16\sqrt{B}L)$ per step on average.

*Proof of Lemma 6.2.* For any $t_s \leq t < m_s$, we can show the following property (See Lemma D.2),

$$F(\mathbf{x}_{t+1}) \leq F(\mathbf{x}_t) - \frac{\eta_t}{2}\|\mathbf{d}_t\|_2^2 + \frac{\eta_t}{2}\|\boldsymbol{\epsilon}_t\|_2^2 + \frac{L}{2}\|\mathbf{x}_{t+1} - \mathbf{x}_t\|_2^2. \tag{6.2}$$

Plugging the update rule $\mathbf{x}_{t+1} = \mathbf{x}_t - \eta_t\mathbf{d}_t$ into (6.2) gives,

$$\begin{aligned}
F(\mathbf{x}_{t+1}) &= F(\mathbf{x}_t) - \|\mathbf{x}_{t+1} - \mathbf{x}_t\|_2^2\left(\frac{1}{2\eta_t} - \frac{L}{2}\right) + \frac{\eta_t\|\boldsymbol{\epsilon}_t\|_2^2}{2} \\
&\leq F(\mathbf{x}_t) - \eta^2\left(\frac{1}{2\eta_t} - \frac{L}{2}\right) + \frac{\eta_t\epsilon^2}{8}, \\
&\leq F(\mathbf{x}_t) - \frac{\eta\epsilon}{8}
\end{aligned}$$

where the first inequality holds due to the fact that $\eta_t = \eta/\|\mathbf{d}_t\|_2$ and $\|\boldsymbol{\epsilon}_t\|_2 \leq \epsilon/2$, and the second inequality is by $\eta_t = \eta/\|\mathbf{d}_t\|_2 \leq \eta/\epsilon \leq 1/(2L)$. $\square$

Following Lemma shows that if $\mathbf{x}_{m_s}$ is a saddle point, then with high probability, the algorithm will break during the Escape phase and set FIND←false. Thus, whenever $\mathbf{x}_{m_s}$ is not a local minima, the algorithm cannot terminate.

**Lemma 6.3.** Under Assumptions 3.1 and 3.2, set perturbation radius $r \leq L\eta_H\epsilon_H/(C\rho)$, step size $\eta_H \leq \min\{1/(16L\log(\eta_H\epsilon_H\sqrt{d}LC^{-1}\rho^{-1}\delta^{-1}r^{-1})), 1/(8CL\log\ell_{\text{thres}})\} = \widetilde{O}(L^{-1})$, $\ell_{\text{thres}} = 2\log(\eta_H\epsilon_H\sqrt{d}LC^{-1}\rho^{-1}\delta^{-1}r^{-1})/(\eta_H\epsilon_H) = \widetilde{O}(\eta_H^{-1}\epsilon_H^{-1})$, and $\overline{D} < C^2L^2\eta_H^2\epsilon_H^2/(\rho^2\ell_{\text{thres}}^2)$, where $C = O(\log(d\ell_{\text{thres}}/\delta) = \widetilde{O}(1)$. We also set $b = q = \sqrt{B} \geq 16\log(4/\delta)/(\eta_H^2\epsilon_H^2)$. Then for any $s$, when $\lambda_{\min}(\nabla^2 F(\mathbf{x}_{m_s})) \leq -\epsilon_H$, with probability at least $1 - 2\delta$ algorithm breaks in the Escape phase.

*Proof of Lemma 6.3.* Let $\{\mathbf{x}_t\}, \{\mathbf{x}_t'\}$ be two coupled sequences by running `Pullback-SPIDER` from $\mathbf{x}_{m_s+1}, \mathbf{x}_{m_s+1}'$ with $\mathbf{x}_{m_s+1} - \mathbf{x}_{m_s+1}' = r_0\mathbf{e}_1$, where $\mathbf{x}_{m_s+1}, \mathbf{x}_{m_s+1}' \in \mathbb{B}_{\mathbf{x}_{m_s}}(r)$. Here $r_0 = \delta r/\sqrt{d}$ and $\mathbf{e}_1$ denotes the smallest eigenvector direction of Hessian $\nabla^2 F(\mathbf{x}_{m_s})$.

When $\lambda_{\min}(\nabla^2 F(\mathbf{x}_{m_s})) \leq -\epsilon_H$, under the parameter choice in Lemma 6.3, we can show that at least one of two sequence will escape the saddle point (See Lemma D.3). To be specific, with probability at least $1 - \delta$,

$$\max_{m_s < t < m_s + \ell_{\text{thres}}} \{\|\mathbf{x}_t - \mathbf{x}_{m_s+1}\|_2, \|\mathbf{x}'_t - \mathbf{x}'_{m_s+1}\|_2\} \geq \frac{L\eta_H \epsilon_H}{C\rho}. \tag{6.3}$$

(6.3) suggests that for any two points $\mathbf{x}_{m_s+1}, \mathbf{x}'_{m_s+1}$ satisfying $\mathbf{x}_{m_s+1} - \mathbf{x}'_{m_s+1} = r_0 \mathbf{e}_1$, at least one of them will generate a sequence of iterates which finally move more than $L\eta_H \epsilon_H/(C\rho)$. Thus, let $\mathcal{S} \subseteq \mathbb{B}_{m_s}(r)$ be the set of $\mathbf{x}_{m_s+1}$ which will not generate a sequence of iterates moving more than $\frac{L\eta_H \epsilon_H}{C\rho}$, then in the direction $\mathbf{e}_1$, the "thickness" of $\mathcal{S}$ is smaller than $r_0$. Simple integration shows that the ratio between the volume of $\mathcal{S}$ and $\mathbb{B}_{m_s}(r)$ is bounded by $\delta$. Therefore, since $\mathbf{x}_{m_s+1}$ is generated from $\mathbf{x}_{m_s}$ by adding a uniform random noise in ball $\mathbb{B}_{m_s}(r)$, we conclude that the probability for $\mathbf{x}_{m_s+1}$ locating in $\mathcal{S}$ is less than $\delta$. Applying union bound, we get with probability at least $1 - 2\delta$,

$$\exists m_s < t < m_s + \ell_{\text{thres}}, \|\mathbf{x}_t - \mathbf{x}_{m_s+1}\|_2 \geq \frac{L\eta_H \epsilon_H}{C\rho}. \tag{6.4}$$

Denote $\mathcal{E}$ as the event that the algorithm does not break in the Escape phase. Then under $\mathcal{E}$, for any $m_s < t < m_s + \ell_{\text{thres}}$, we have

$$\|\mathbf{x}_t - \mathbf{x}_{m_s+1}\|_2 \leq \sum_{i=m_s+1}^{t-1} \|\mathbf{x}_{i+1} - \mathbf{x}_i\|_2 \leq \sqrt{(t - m_s)\sum_{i=m_s}^{t-1} \|\mathbf{x}_{i+1} - \mathbf{x}_i\|_2^2} \leq (t - m_s)\sqrt{\overline{D}},$$

where the first inequality is due to the triangle inequality and the second inequality is due to Cauchy-Schwarz inequality. Thus, by the choice of $\ell_{\text{thres}}$ and $\overline{D}$, we have

$$\|\mathbf{x}_t - \mathbf{x}_{m_s+1}\|_2 \leq (t - m_s)\sqrt{\overline{D}} \leq \ell_{\text{thres}}\sqrt{\overline{D}} < C \cdot \frac{L\eta_H \epsilon_H}{\rho}.$$

Then by (6.4), we know that $\mathbb{P}(\mathcal{E}) \leq 2\delta$. Therefore when $\lambda_{\min}(\nabla^2 F(\mathbf{x}_{m_s})) \leq -\epsilon_H$, with probability at least $1 - 2\delta$, `Pullback` breaks in the Escape phase. □

Next lemma bounds the decreasing value of the function during the Escape phase if the algorithm breaks in the Escape phase(i.e. FIND is false).

**Lemma 6.4** (localization). Suppose the result of Lemma 6.1 holds, and set the step size $\eta_H \leq 1/\left(L\sqrt{128\log(4/\delta)}\right)$, perturbation radius $r \leq \min\left\{\log(4/\delta)\eta_H\sigma^2/(2B\epsilon), \sqrt{2\log(4/\delta)\eta_H\sigma^2/(BL)}\right\}$, and $\overline{D} = \sigma^2/(4BL^2)$. Suppose the algorithm breaks in the Escape phase starting at $\mathbf{x}_{m_s}$, then we have

$$F(\mathbf{x}_{m_s}) - F(\mathbf{x}_{t_{s+1}}) \geq (t_{s+1} - m_s)\frac{\log(4/\delta)\eta_H\sigma^2}{B}.$$

*Proof of Lemma 6.4.* For any $m_s < i < t_{s+1}$, we can show the following property (See Lemma D.2),

$$F(\mathbf{x}_{i+1}) \leq F(\mathbf{x}_i) - \frac{\eta_i}{2}\|\mathbf{d}_i\|_2^2 + \frac{\eta_i}{2}\|\boldsymbol{\epsilon}_i\|_2^2 + \frac{L}{2}\|\mathbf{x}_{i+1} - \mathbf{x}_i\|_2^2. \tag{6.5}$$

Plugging the update rule $\mathbf{x}_{i+1} = \mathbf{x}_i - \eta_i \mathbf{d}_i$ into (6.5) gives,

$$F(\mathbf{x}_{i+1}) \leq F(\mathbf{x}_i) + \frac{\eta_i}{2}\|\boldsymbol{\epsilon}_i\|_2^2 - \left(\frac{1}{2\eta_i} - \frac{L}{2}\right)\|\mathbf{x}_{i+1} - \mathbf{x}_i\|_2^2$$

$$\leq F(\mathbf{x}_i) + \frac{\eta_H}{2}\frac{8\log(4/\delta)\sigma^2}{B} - \frac{1}{4\eta_H}\|\mathbf{x}_{i+1} - \mathbf{x}_i\|_2^2 \tag{6.6}$$

where the the second inequality holds due to Lemma 6.1 and $\eta_i \leq \eta_H \leq 1/(2L)$ for any $m_s < i < t_{s+1}$. Telescoping (6.6) from $i = m_s + 1$ to $t_{s+1} - 1$, we have

$$F(\mathbf{x}_{t_{s+1}}) \leq F(\mathbf{x}_{m_s+1}) + 4\eta_H \log(4/\delta)(t_{s+1} - m_s - 1)\frac{\sigma^2}{B} - \frac{1}{4\eta_H}\sum_{i=m_s+1}^{t_{s+1}-1} \|\mathbf{x}_{i+1} - \mathbf{x}_i\|_2^2.$$

Finally, we have

$$
\begin{aligned}
F(\mathbf{x}_{m_s+1}) - F(\mathbf{x}_{t_{s+1}}) &\geq \sum_{i=m_s+1}^{t_{s+1}-1} \frac{\|\mathbf{x}_{i+1} - \mathbf{x}_i\|_2^2}{4\eta_H} - 4\log(4/\delta)(t_{s+1} - m_s - 1)\eta_H \frac{\sigma^2}{B} \\
&= (t_{s+1} - m_s - 1)\left( \frac{\overline{D}}{4\eta_H} - \frac{4\log(4/\delta)\eta_H\sigma^2}{B} \right) \\
&= (t_{s+1} - m_s - 1)\left( \frac{\sigma^2}{16\eta_H B L^2} - \frac{4\log(4/\delta)\eta_H\sigma^2}{B} \right) \\
&\geq (t_{s+1} - m_s - 1)\frac{4\log(4/\delta)\eta_H\sigma^2}{B},
\end{aligned}
\tag{6.7}
$$

where the last inequality is by the choice of $\eta_H \leq 1/\left(L\sqrt{128\log(4/\delta)}\right)$. For $i = m_s$, we have (See Lemma D.2)

$$
F(\mathbf{x}_{m_s+1}) \leq F(\mathbf{x}_{m_s}) + (\|\mathbf{d}_{m_s}\|_2 + \|\boldsymbol{\epsilon}_{m_s}\|_2 + Lr/2)r.
\tag{6.8}
$$

Plugging $\|\mathbf{d}_{m_s}\|_2 \leq \epsilon$ and $\|\boldsymbol{\epsilon}_{m_s}\|_2 \leq \epsilon/2$ into (6.8) gives,

$$
F(\mathbf{x}_{m_s+1}) \leq F(\mathbf{x}_{m_s}) + (4\epsilon + Lr/2)r \leq F(\mathbf{x}_{m_s}) + \frac{2\log(4/\delta)\eta_H\sigma^2}{B},
\tag{6.9}
$$

where the last inequality is by the choice $r \leq \min\left\{ \log(4/\delta)\eta_H\sigma^2/(2B\epsilon), \sqrt{2\log(4/\delta)\eta_H\sigma^2/(BL)} \right\}$. Combining (6.7) and (6.9) and applying $t_{s+1} - m_s \geq 2$ gives,

$$
F(\mathbf{x}_{m_s}) - F(\mathbf{x}_{t_{s+1}}) \geq [4(t_{s+1} - m_s - 1) - 2]\frac{\log(4/\delta)\eta_H\sigma^2}{B} \geq (t_{s+1} - m_s)\frac{\log(4/\delta)\eta_H\sigma^2}{B}.
$$

$\square$

Now, we can provide the proof of Theorem 5.1 .

*Proof of Theorem 5.1.* The analysis can be divided into two phases, i.e., GD phase and Escape phase. The function value will decrease at different rates in different phases.

**GD phase**: In this phase, $\|\mathbf{d}_t\|_2 \geq \epsilon$ and $\|\boldsymbol{\epsilon}\|_2 \leq \epsilon/2$ due to Lemma 6.1. Thus the gradients of the function are large $\|\nabla F(\mathbf{x})\|_2 \geq \epsilon/2$. Lemma 6.2 further shows that the loss decreases by at least $\sigma\epsilon/(16\sqrt{B}L)$ on average.

**Escape phase:** In this phase, the starting point $\mathbf{x}_{m_s}$ satisfies $\|\nabla F(\mathbf{x}_{m_s})\|_2 \leq \|\mathbf{d}_{m_s}\|_2 + \|\boldsymbol{\epsilon}_t\|_2 \leq 2\epsilon$. If $\mathbf{x}_{m_s}$ is a saddle point with $\lambda_{\min}(\nabla^2 F(\mathbf{x}_{m_s})) \leq -\epsilon_H$, then by Lemma 6.3, with high probability `Pullback-SPIDER` will break Escape phase, set FIND←False and begin a new GD phase. Further by Lemma 6.4, the loss will decrease by at least $\log(4/\delta)\eta_H\sigma^2/B$ per step on average.

**Sample Complexity:** Note that the total amount for function value can decrease is at most $\Delta = F(\mathbf{x}_0) - \inf_\mathbf{x} F(\mathbf{x}) < +\infty$. So the algorithm must end and find an $(\epsilon, \epsilon_H)$-approximate local minimum within $\widetilde{O}(\sqrt{B}L\Delta\sigma^{-1}\epsilon^{-1} + BL\Delta\sigma^{-2})$ iterations. Notice that on average we sample $\max\{b, B/q\} = \sqrt{B}$ examples per iteration, so the total sample complexity is $\widetilde{O}(BL\Delta\sigma^{-1}\epsilon^{-1} + B^{3/2}L\Delta\sigma^{-2})$. Plugging in the choice of $B = \widetilde{O}(\sigma^2\epsilon^{-2} + \sigma^2\rho^2\epsilon_H^{-4})$ in Theorem 5.1, we have the total gradient complexity

$$
\widetilde{O}\left( \frac{\sigma L\Delta}{\epsilon^3} + \frac{\sigma\rho^2 L\Delta}{\epsilon\epsilon_H^4} + \frac{\sigma\rho^3 L\Delta}{\epsilon_H^6} \right) = \widetilde{O}\left( \frac{\sigma L\Delta}{\epsilon^3} + \frac{\sigma\rho^3 L\Delta}{\epsilon_H^6} \right),
$$

where the equation is due to the Young's inequality.

$\square$

# 7 Conclusions

In this paper, we propose a perturbed stochastic gradient framework named `Pullback` for finding local minima. `Pullback` can find $(\epsilon, \epsilon_H)$-approximate local minima within $\widetilde{O}(\epsilon^{-3} + \epsilon_H^{-6})$ stochastic gradient evaluations, which matches the best possible complexity results in the classical $\epsilon_H = \sqrt{\epsilon}$ setting. Our results show that simple perturbed gradient methods can be as efficient as more sophisticated algorithms for finding local minima.

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
