## A    Proof of Theorem 5.3

In this section we present the main proof to Theorem 5.3. We define $\boldsymbol{\epsilon}_t = \mathbf{d}_t - \nabla F(\mathbf{x}_t)$ for simplicity.

To prove the main theorem, we need two groups of lemmas to charctrize the behavior of the Algorithm `Pullback-STORM`.

Next lemma provides the upper bound of $\boldsymbol{\epsilon}_t$.

**Lemma A.1.** Set $\eta \leq \sigma/(2bL)$, $r \leq \sigma/(2bL)$ and $\overline{D} \leq \sigma^2/(4b^2L^2)$, $a = 56^2 \log(4/\delta)/b$, $B = b^2$, $a \leq 1/4\ell_{\text{thres}}$ , with probability at least $1 - 2\delta$, for all $t$ we have

$$\|\boldsymbol{\epsilon}_t\|_2 \leq \frac{2^{10} \log(4/\delta)\sigma}{b}.$$

Furthermore, by the choice of $b$ in Theorem 5.1 we have that $\|\boldsymbol{\epsilon}_t\|_2 \leq \epsilon/2$.

*Proof.* See Appendix B.1. □

**Lemma A.2.** Suppose the event in Lemma A.1 holds and $\eta \leq \epsilon/(2L)$, then for any $s$, we have

$$F(\mathbf{x}_{t_s}) - F(\mathbf{x}_{m_s}) \geq \frac{(m_s - t_s)\eta\epsilon}{8}.$$

*Proof.* The proof is the same as that of Lemma 6.2, with the fact $\|\boldsymbol{\epsilon}_t\|_2 \leq \epsilon/2$ from Lemma A.1. □

The choice of $\eta$ in Theorem 5.3 further implies that the loss decrease by $\sigma\epsilon/(16bL)$ on average.

Next lemma shows that if $\mathbf{x}_{m_s}$ is a saddle point, then with high probability, the algorithm will break during the Escape phase and set FIND←false. Thus, whenever $\mathbf{x}_{m_s}$ is not a local minimum, the algorithm cannot terminate.

**Lemma A.3.** Under Assumptions 3.1 and 3.2, set $r \leq L\eta_H\epsilon_H/\rho$, $a \leq \eta_H\epsilon_H$, $b \geq \max\{16\log(4/\delta)\eta_H^{-2}L^{-2}\epsilon_H^{-2}, 56^2 \log(4/\delta)a^{-1}\}$, $\ell_{\text{thres}} = 2\log(8\epsilon_H\sqrt{d}\rho^{-1}\delta^{-1}r^{-1})/(\eta_H\epsilon_H)$, $\eta_H \leq \min\{1/(10L\log(8\epsilon_HL\rho^{-1}r_0^{-1})), 1/(10L\log(\ell_{\text{thres}}))\}$ and $\overline{D} < L^2\eta_H^2\epsilon_H^2/(\rho\ell_{\text{thres}}^2)$. Then for any $s$, when $\lambda_{\min}(\nabla^2 F(\mathbf{x}_{m_s})) \leq -\epsilon_H$, with probability at least $1 - 2\delta$ algorithm breaks in the Escape phase.

*Proof.* See Appendix B.2. □

Next lemma shows that `Pullback-STORM` decreases when it breaks.

**Lemma A.4** (localization). Suppose the event in Lemma A.1 holds, and $r \leq \min\{\log(4/\delta)^2\eta_H\sigma^2/(4b^2\epsilon), \sqrt{2\log(4/\delta)^2\eta_H\sigma^2/(b^2L)}\}$, $\eta_H \leq 1/(2^{12}L\log(4/\delta))$, $\overline{D} = \sigma^2/(4b^2L^2)$. Then for any $s$, when `Pullback-STORM` breaks, then $\mathbf{x}_{m_s}$ satisfies

$$F(\mathbf{x}_{m_s}) - F(\mathbf{x}_{t_{s+1}}) \geq (t_{s+1} - m_s)\frac{\log(4/\delta)^2\eta_H\sigma^2}{b^2}. \tag{A.1}$$

*Proof.* See Appendix B.3. □

With all above lemmas, we prove Theorem 5.3.

*Proof of Theorem 5.3.* Under the choice of parameter in Theorem 5.3, we have Lemma A.1 to A.4 hold. Now for GD phase, we know that the function value $F$ decreases by $\sigma\epsilon/(16bL)$ on average. For Escape phase, we know that the $F$ decreases by $\log(4/\delta)^2\eta_H\sigma^2/b^2$ on average. So `Pullback-STORM` can find $(\epsilon, \epsilon_H)$-approximate local minima within $\widetilde{O}(bL\Delta\sigma^{-1}\epsilon^{-1} + b^2L\Delta\sigma^{-2})$ iterations (we use the fact that $\eta_H = \widetilde{O}(L^{-1})$). Then the total number of stochastic gradient evaluations is bounded by $\widetilde{O}(B + b^2L\Delta\sigma^{-1}\epsilon^{-1} + b^3L\Delta\sigma^{-2})$. Plugging in the choice of $b = \widetilde{O}(\sigma\epsilon^{-1} + \sigma\rho\epsilon_H^{-2})$ in Theorem 5.3, we have the total sample complexity

$$\widetilde{O}\left(\frac{\sigma L\Delta}{\epsilon^3} + \frac{\sigma\rho^2 L\Delta}{\epsilon\epsilon_H^4} + \frac{\sigma\rho^3 L\Delta}{\epsilon_H^6}\right).$$

The proof finishes by using Young's inequality. □

# B Proof of Lemmas in Section A

In this section we prove lemmas in Section A. Let filtration $\mathcal{F}_{t,b}$ denote the all history before sample $\boldsymbol{\xi}_{t,b}$ at time $t \in \{0, \cdots, T\}$, then it is obvious that $\mathcal{F}_{0,1} \subseteq \mathcal{F}_{0,b} \subseteq \cdots \subseteq \mathcal{F}_{1,1} \subseteq \cdots \subseteq \mathcal{F}_{T,1} \subseteq \cdots \subseteq \mathcal{F}_{T,b}$.

We also need the following fact:

**Proposition B.1.** For any $t$, we have the following equation:

$$\frac{\boldsymbol{\epsilon}_{t+1}}{(1-a)^{t+1}} - \frac{\boldsymbol{\epsilon}_t}{(1-a)^t} = \frac{1}{(1-a)^{t+1}} \sum_{i \leq b} \boldsymbol{\epsilon}_{t,i},$$

where

$$\boldsymbol{\epsilon}_{t,i} = \frac{a}{b} [\nabla f(\mathbf{x}_{t+1}; \boldsymbol{\xi}_{t+1}^i) - \nabla F(\mathbf{x}_{t+1})]$$
$$+ \frac{1-a}{b} [\nabla F(\mathbf{x}_t) - \nabla f(\mathbf{x}_t; \boldsymbol{\xi}_{t+1}^i) - \nabla F(\mathbf{x}_{t+1}) + \nabla f(\mathbf{x}_{t+1}; \boldsymbol{\xi}_{t+1}^i)].$$

*Proof.* Following the update rule in `Pullback-STORM`, we could have the update rule of $\boldsymbol{\epsilon}$ described as

$$\boldsymbol{\epsilon}_{t+1} = \frac{1-a}{b} \sum_{i \leq b} [\mathbf{d}_t - \nabla f(\mathbf{x}_t; \boldsymbol{\xi}_{t+1}^i)] + \frac{1}{b} \sum_{i \leq b} [\nabla f(\mathbf{x}_{t+1}; \boldsymbol{\xi}_{t+1}^i) - \nabla F(\mathbf{x}_{t+1})]$$

$$= \frac{a}{b} \sum_{i \leq b} [\nabla f(\mathbf{x}_{t+1}; \boldsymbol{\xi}_{t+1}^i) - \nabla F(\mathbf{x}_{t+1})] + (1-a)(\mathbf{d}_t - \nabla F(\mathbf{x}_t))$$

$$+ \frac{1-a}{b} \sum_{i \leq b} [\nabla F(\mathbf{x}_t) - \nabla f(\mathbf{x}_t; \boldsymbol{\xi}_{t+1}^i) - \nabla F(\mathbf{x}_{t+1}) + \nabla f(\mathbf{x}_{t+1}; \boldsymbol{\xi}_{t+1}^i)]$$

$$= \frac{a}{b} \sum_{i \leq b} [\nabla f(\mathbf{x}_{t+1}; \boldsymbol{\xi}_{t+1}^i) - \nabla F(\mathbf{x}_{t+1})] + (1-a)\boldsymbol{\epsilon}_t$$

$$+ \frac{1-a}{b} \sum_{i \leq b} [\nabla F(\mathbf{x}_t) - \nabla f(\mathbf{x}_t; \boldsymbol{\xi}_{t+1}^i) - \nabla F(\mathbf{x}_{t+1}) + \nabla f(\mathbf{x}_{t+1}; \boldsymbol{\xi}_{t+1}^i)],$$

where the last equation is by definition $\boldsymbol{\epsilon}_t := \mathbf{d}_t - \nabla F(\mathbf{x}_t)$. Thus we have

$$\frac{\boldsymbol{\epsilon}_{t+1}}{(1-a)^{t+1}} - \frac{\boldsymbol{\epsilon}_t}{(1-a)^t}$$

$$= \frac{1}{(1-a)^{t+1}} \left( \frac{a}{b} \sum_{i \leq b} [\nabla f(\mathbf{x}_{t+1}; \boldsymbol{\xi}_{t+1}^i) - \nabla F(\mathbf{x}_{t+1})] \right.$$

$$\left. + \frac{1-a}{b} \sum_{i \leq b} [\nabla F(\mathbf{x}_t) - \nabla f(\mathbf{x}_t; \boldsymbol{\xi}_{t+1}^i) - \nabla F(\mathbf{x}_{t+1}) + \nabla f(\mathbf{x}_{t+1}; \boldsymbol{\xi}_{t+1}^i)] \right),$$

$$= \frac{1}{(1-a)^{t+1}} \sum_{i \leq b} \boldsymbol{\epsilon}_{t,i}.$$

$\square$

## B.1 Proof of Lemma A.1

**Proposition B.2.** For two positive sequences $\{a_i\}_{i=1}^n$ and $\{b_i\}_{i=1}^n$. Suppose $C = \max_{i,j \in [n]} \{|a_i/a_j|\}, \bar{b} = \sum_{i=1}^n b_i/n$. Then we have,

$$\sum_{i=1}^n a_i b_i \leq \max_i a_i \cdot n \cdot \bar{b} \leq C \sum_{i=1}^n a_i \bar{b}.$$

*Proof of Lemma A.1.* By Proposition B.1 we have

$$\frac{\boldsymbol{\epsilon}_{t+1}}{(1-a)^{t+1}} - \frac{\boldsymbol{\epsilon}_t}{(1-a)^t} = \frac{1}{(1-a)^{t+1}} \sum_{i \leq b} \boldsymbol{\epsilon}_{t,i}.$$

It is easy to verify that $\{\boldsymbol{\epsilon}_{t,i}\}$ forms a martingale difference sequence and

$$
\begin{aligned}
\|\boldsymbol{\epsilon}_{t,i}\|_2^2 &\leq 2\left\|\frac{a}{b}\big[\nabla f(\mathbf{x}_{t+1};\boldsymbol{\xi}_{t+1}^i) - \nabla F(\mathbf{x}_{t+1})\big]\right\|_2^2 \\
&\quad + 2\left\|\frac{1-a}{b}\big[\nabla F(\mathbf{x}_t) - \nabla f(\mathbf{x}_t;\boldsymbol{\xi}_{t+1}^i) - \nabla F(\mathbf{x}_{t+1}) + \nabla f(\mathbf{x}_{t+1};\boldsymbol{\xi}_{t+1}^i)\big]\right\|_2^2 \\
&\leq \frac{2a^2\sigma^2 + 8(1-a)^2 L^2 \|\mathbf{x}_{t+1} - \mathbf{x}_i\|_2^2}{b^2},
\end{aligned}
$$

where the first inequality holds due to triangle inequality, the second inequality holds due to Assumptions 3.1 and 3.2. Therefore, by Azuma-Hoeffding inequality (See Lemma D.1 for detail), with probability at least $1 - \delta$, we have that for any $t > 0$,

$$
\begin{aligned}
\left\|\frac{\boldsymbol{\epsilon}_t}{(1-a)^t} - \frac{\boldsymbol{\epsilon}_0}{(1-a)^0}\right\|_2^2 &\leq 4\log(4/\delta) \sum_{i=0}^{t-1} b \cdot \frac{2a^2\sigma^2 + 8(1-a)^2 L^2 \|\mathbf{x}_{i+1} - \mathbf{x}_i\|_2^2}{(1-a)^{2i+2} b^2} \\
&= 8\log(4/\delta) \sum_{i=0}^{t-1} \frac{a^2\sigma^2 + 4(1-a)^2 L^2 \|\mathbf{x}_{i+1} - \mathbf{x}_i\|_2^2}{(1-a)^{2i+2} b}.
\end{aligned}
$$

Therefore, we have

$$
\begin{aligned}
\|\boldsymbol{\epsilon}_t\|_2^2 &\leq 2(1-a)^{2t}\left\|\frac{\boldsymbol{\epsilon}_t}{(1-a)^t} - \boldsymbol{\epsilon}_0\right\|_2^2 + 2(1-a)^{2t}\|\boldsymbol{\epsilon}_0\|_2^2 \\
&\leq \log(4/\delta)\left[\frac{64L^2}{b} \sum_{i=0}^{t-1}(1-a)^{2t-2i}\|\mathbf{x}_{i+1} - \mathbf{x}_i\|_2^2 + \frac{16a\sigma^2}{b}\right] + 2(1-a)^{2t}\|\boldsymbol{\epsilon}_0\|_2^2. \quad \text{(B.1)}
\end{aligned}
$$

By Azuma-Hoeffding Inequality, we have with probability $1 - \delta$,

$$
\|\boldsymbol{\epsilon}_0\|_2^2 = \left\|\frac{1}{B}\sum_{1\leq i\leq B}\big[\nabla f(\mathbf{x}_0;\boldsymbol{\xi}_0^i) - \nabla F(\mathbf{x}_0)\big]\right\|_2^2 \leq \frac{4\log(4/\delta)\sigma^2}{B}.
$$

Therefore, with probability $1 - 2\delta$, we have

$$
\begin{aligned}
\|\boldsymbol{\epsilon}_t\|_2^2 &\leq \log(4/\delta)\left[\frac{64L^2}{b}\sum_{i=0}^{t-1}(1-a)^{2t-2i-2}\|\mathbf{x}_{i+1} - \mathbf{x}_i\|_2^2 + \frac{16a\sigma^2}{b} + \frac{32(1-a)^{2t}\sigma^2}{B}\right] \\
&= \frac{64L^2\log(4/\delta)}{b}\underbrace{\sum_{i=0}^{t-1}(1-a)^{2t-2i-2}\|\mathbf{x}_{i+1} - \mathbf{x}_i\|_2^2}_{I} + \frac{16a\sigma^2\log(4/\delta)}{b} \quad \text{(B.2)} \\
&\quad + \frac{32(1-a)^{2t}\log(4/\delta)\sigma^2}{B}. \quad \text{(B.3)}
\end{aligned}
$$

We now bound $I$. Denote $S_1 = \{i \in [t-1] | \exists j, t_j \leq i < m_j\}$, $S_2 = \{i \in [t-1] | \exists j, i = m_j\}$, $S_3 = \{i \in [t-1] | \exists j, m_j < i < t_{j+1}\}$, We can divide $I$ into three part,

$$
\begin{aligned}
I &= \underbrace{\sum_{i\in S_1}(1-a)^{2t-2i-2}\|\mathbf{x}_{i+1} - \mathbf{x}_i\|_2^2}_{I_1} + \underbrace{\sum_{i\in S_2}(1-a)^{2t-2i-2}\|\mathbf{x}_{i+1} - \mathbf{x}_i\|_2^2}_{I_2} \\
&\quad + \underbrace{\sum_{i\in S_3}^{t-1}(1-a)^{2t-2i-2}\|\mathbf{x}_{i+1} - \mathbf{x}_i\|_2^2}_{I_3}. \quad \text{(B.4)}
\end{aligned}
$$

Because $\|\mathbf{x}_{i+1} - \mathbf{x}_i\|_2 = \eta_t \|\mathbf{d}_i\|_2 = \eta$, we can bound $I_1$ as follows,

$$
I_1 = \eta^2 \sum_{i\in S_1}(1-a)^{2t-2i-2} \leq \eta^2 \sum_{i=0}^{\infty}(1-a)^i = \frac{\eta^2}{a}. \quad \text{(B.5)}
$$

Because the perturbation radius is $r$, we can bound $I_2$ as follows,

$$I_2 = \sum_{i \in S_2} (1-a)^{2t-2i-2} \|\mathbf{x}_{i+1} - \mathbf{x}_i\|_2^2 \le r^2 \sum_{i \in S_2} (1-a)^{2t-2i-2} \le \frac{r^2}{a}. \tag{B.6}$$

To bound $I_3$, we have

$$
\begin{aligned}
I_3 &= \sum_{i \in S_3}^{t-1} (1-a)^{2t-2i-2} \|\mathbf{x}_{i+1} - \mathbf{x}_i\|_2^2 \\
&= \sum_{s=1}^{S} \sum_{i=m_s+1}^{\min\{t-1,t_{s+1}-1\}} (1-a)^{2t-2i-2} \|\mathbf{x}_{i+1} - \mathbf{x}_i\|_2^2 \\
&\le \sum_{s=1}^{S} (1-a)^{-2\ell_{\text{thres}}} \sum_{i=m_s+1}^{\min\{t-1,t_{s+1}-1\}} (1-a)^{2t-2i-2} \overline{D} \\
&= (1-a)^{-2\ell_{\text{thres}}} \sum_{i \in S_3}^{t-1} (1-a)^{2t-2i-2} \overline{D} \\
&\le \frac{\overline{D}(1-a)^{-2\ell_{\text{thres}}}}{a}, \tag{B.7}
\end{aligned}
$$

where $S$ satisfies $m_S < t-1 < t_{S+1}$. The first inequality holds due to Proposition B.2 with the fact that the average of $\|\mathbf{x}_{i+1} - \mathbf{x}_i\|_2^2$ is bounded by $\overline{D}$, according to the `Pullback` scheme, and $t_{s+1} - m_s < \ell_{\text{thres}}$, the last one holds trivially. Substituting (B.5), (B.6), (B.7) into (B.4), we have

$$I \le \frac{\eta^2 + r^2 + (1-a)^{2\ell_{\text{thres}}} \overline{D}}{a}.$$

Therefore (B.3) can further bounded by

$$\|\boldsymbol{\epsilon}_t\|_2^2 \le \frac{64L^2 \log(4/\delta)}{b} \frac{\eta^2 + r^2 + (1-a)^{2\ell_{\text{thres}}} \overline{D}}{a} + \frac{16a\sigma^2 \log(4/\delta)}{b} + \frac{32(1-a)^{2t} \log(4/\delta)\sigma^2}{B}. \tag{B.8}$$

By the selection of $\eta \le \sigma/(2bL)$, $r \le \sigma/(2bL)$ and $\overline{D} \le \sigma^2/(4b^2 L^2)$, $a = 56^2 \log(4/\delta)/b$, $B = b^2$, $a \le 1/4\ell_{\text{thres}}$, it's easy to verify that

$$\frac{64L^2 \log(4/\delta)}{b} \frac{\eta^2 + r^2 + 2\overline{D}}{a} \le \frac{\sigma^2}{b^2} \tag{B.9}$$

$$(1-a)^{2\ell_{\text{thres}}} \ge 1 - 2a\ell_{\text{thres}} \ge \frac{1}{2} \tag{B.10}$$

$$\frac{16a\sigma^2 \log(4/\delta)}{b} \le \frac{224^2 \sigma^2 \log(4/\delta)^2}{b^2} \tag{B.11}$$

$$\frac{32 \log(4/\delta)\sigma^2}{B} \le \frac{32 \log(4/\delta)\sigma^2}{b^2}. \tag{B.12}$$

Plugging (B.9) to (B.12) into (B.8) gives,

$$\|\boldsymbol{\epsilon}_t\|_2 \le \frac{2^{10} \log(4/\delta)\sigma}{b}.$$

$\square$

## B.2 Proof of Lemma A.3

**Lemma B.3** (Small stuck region). Suppose $-\gamma = \lambda_{\min}(\nabla^2 F(\mathbf{x}_{m_s})) \le -\epsilon_H$. Set $\ell = 2\log(8\epsilon_H \rho^{-1} r_0^{-1})/(\eta_H \gamma)$, $\eta_H \le \min\{1/(10L \log(8\epsilon_H L\rho^{-1} r_0^{-1})), 1/(10L \log(\ell))\}$, $a \le \eta_H \gamma$, $r \le L\eta_H \epsilon_H/\rho$. Let $\{\mathbf{x}_t\}, \{\mathbf{x}_t'\}$ be two coupled sequences by running `Pullback-STORM` from $\mathbf{x}_{m_s+1}, \mathbf{x}_{m_s+1}'$ with $\mathbf{w}_{m_s+1} = \mathbf{x}_{m_s+1} - \mathbf{x}_{m_s+1}' = r_0 \mathbf{e}_1$, where $\mathbf{x}_{m_s+1}, \mathbf{x}_{m_s+1}' \in \mathbb{B}_{\mathbf{x}_{m_s}}(r)$,

$r_0 = \delta r / \sqrt{d}$ and $\mathbf{e}_1$ denotes the smallest eigenvector direction of Hessian $\nabla^2 F(\mathbf{x}_{m_s})$. Moreover, let batch size $b \geq \max\{16 \log(4/\delta)\eta_H^{-2}L^{-2}\gamma^{-2}, 56^2 \log(4/\delta)a^{-1}\}$, then with probability $1 - 2\delta$ we have

$$\exists T \leq \ell, \max\{\|\mathbf{x}_T - \mathbf{x}_0\|_2, \|\mathbf{x}'_T - \mathbf{x}'_0\|_2\} \geq \frac{\eta_H \epsilon_H L}{\rho}.$$

*Proof.* See Appendix C.1. □

*Proof of Lemma A.3.* We assume $\lambda_{\min}(\nabla^2 F(\mathbf{x}_{m_s})) < -\epsilon_H$ and prove our statement by contradiction. Lemma B.3 shows that, in the random perturbation ball at least one of two points in the $\mathbf{e}_1$ direction will escape the saddle point if their distance is larger than $r_0 = \frac{\delta r}{\sqrt{d}}$. Thus, the probability of the starting point $\mathbf{x}_{m_s+1} \sim \mathbb{B}_{\mathbf{x}_{m_s}}(r)$ located in the stuck region uniformly is less than $\delta$. Then with probability at least $1 - 2\delta$,

$$\exists m_s < t < m_s + \ell_{\text{thres}}, \|\mathbf{x}_t - \mathbf{x}_{m_s}\|_2 \geq \frac{L\eta_H \epsilon_H}{\rho}. \tag{B.13}$$

Suppose `Pullback-STORM` does not break, then for any $m_s < t < m_s + \ell_{\text{thres}}$,

$$\|\mathbf{x}_t - \mathbf{x}_{m_s}\|_2 \leq \sum_{i=m_s}^{t-1} \|\mathbf{x}_{i+1} - \mathbf{x}_i\|_2 \leq \sqrt{(t-m_s)\sum_{i=m_s}^{t-1} \|\mathbf{x}_{i+1} - \mathbf{x}_i\|_2^2} \leq (t-m_s)\sqrt{\overline{D}},$$

where the first inequality is due to the triangle inequality and the second inequality is due to Cauchy-Schwarz inequality. Thus, by the selection of $\overline{D}$, we have

$$\|\mathbf{x}_t - \mathbf{x}_{m_s}\|_2 \leq (t-m_s)\sqrt{\overline{D}} \leq \ell_{\text{thres}}\sqrt{\overline{D}} < \frac{L\eta_H \epsilon_H}{\rho},$$

which contradicts (B.13). Therefore, we know that with probability at least $1 - 2\delta$, $\lambda_{\min}(\nabla^2 F(\mathbf{x}_{m_s})) \geq -\epsilon_H$. □

## B.3  Proof of Lemma A.4

*Proof of Lemma A.4.* Suppose $m_s < i < t_{s+1}$. Then with probability at least $1 - \delta$, then by Lemma D.2 we have

$$F(\mathbf{x}_{i+1}) \leq F(\mathbf{x}_i) + \frac{\eta_i}{2}\|\boldsymbol{\epsilon}_i\|_2^2 - \left(\frac{1}{2\eta_i} - \frac{L}{2}\right)\|\mathbf{x}_{i+1} - \mathbf{x}_i\|_2^2$$

$$\leq F(\mathbf{x}_i) + \frac{\eta_H}{2}\frac{2^{20}\log(4/\delta)^2\sigma^2}{b^2} - \frac{1}{4\eta_H}\|\mathbf{x}_{i+1} - \mathbf{x}_i\|_2^2 \tag{B.14}$$

where the the second inequality holds due to Lemma A.1 and the fact that for any $m_s < i < t_{s+1}$, $\eta_i \leq \eta_H \leq 1/(2L)$. Taking summation of (B.14) from $i = m_s + 1$ to $t - 1$, we have

$$F(\mathbf{x}_t) \leq F(\mathbf{x}_{m_s+1}) + 2^{19}\eta_H \log(4/\delta)^2(t - m_s - 1)\frac{\sigma^2}{b^2} - \frac{1}{4\eta_H}\sum_{i=m_s+1}^{t-1} \|\mathbf{x}_{i+1} - \mathbf{x}_i\|_2^2. \tag{B.15}$$

Finally, we have

$$\begin{aligned}
F(\mathbf{x}_{m_s+1}) - F(\mathbf{x}_{t_{s+1}}) &\geq \sum_{i=m_s+1}^{t_{s+1}-1} \frac{\|\mathbf{x}_{i+1} - \mathbf{x}_i\|_2^2}{4\eta_H} - 2^{19}\log(4/\delta)^2(t - m_s - 1)\eta_H\frac{\sigma^2}{b^2} \\
&= (t_{s+1} - m_s - 1)\left(\frac{\overline{D}}{4\eta_H} - \frac{2^{19}\log(4/\delta)^2\eta_H\sigma^2}{b^2}\right) \\
&= (t_{s+1} - m_s - 1)\left(\frac{\sigma^2}{16\eta_H b^2 L^2} - \frac{2^{19}\log(4/\delta)^2\eta_H\sigma^2}{b^2}\right) \\
&\geq (t_{s+1} - m_s - 1)\frac{4\log(4/\delta)^2\eta_H\sigma^2}{b^2}, \tag{B.16}
\end{aligned}$$

where the last inequality is by the selection of $\eta_H \leq 1/(2^{12}L\log(4/\delta))$. For $i = m_s$, by Lemma D.2 we have

$$
\begin{aligned}
F(\mathbf{x}_{m_s+1}) &\leq F(\mathbf{x}_t) + (2\|\mathbf{d}_t\|_2 + 2\|\boldsymbol{\epsilon}_t\|_2 + Lr/2)r \\
&\leq F(\mathbf{x}_{m_s}) + (4\epsilon + Lr/2)r \\
&\leq F(\mathbf{x}_{m_s}) + \frac{2\log(4/\delta)^2\eta_H\sigma^2}{b^2},
\end{aligned}
\tag{B.17}
$$

where the last inequality is by the selection, $r \leq \min\left\{\log(4/\delta)^2\eta_H\sigma^2/(4b^2\epsilon), \sqrt{2\log(4/\delta)^2\eta_H\sigma^2/(b^2L)}\right\}$.

Combining (B.16) and (B.17) we have that

$$
\begin{aligned}
F(\mathbf{x}_{m_s}) - F(\mathbf{x}_{t_{s+1}}) &= F(\mathbf{x}_{m_s}) - F(\mathbf{x}_{m_s+1}) + F(\mathbf{x}_{m_s+1}) - F(\mathbf{x}_{t_{s+1}}) \\
&\geq (t_{s+1} - m_s - 1)\frac{4\log(4/\delta)^2\eta_H\sigma^2}{b^2} - \frac{2\log(4/\delta)^2\eta_H\sigma^2}{b^2} \\
&\geq (t_{s+1} - m_s)\frac{\log(4/\delta)^2\eta_H\sigma^2}{b^2},
\end{aligned}
$$

where we use the fact that $t_{s+1} - m_s \geq 2$. $\qquad\square$

## C  Proof of Lemmas in Section B

### C.1  Proof of Lemma B.3

Define $\mathbf{w}_t := \mathbf{x}_t - \mathbf{x}_t'$ as the distance between the two coupled sequences. By the construction, we have that $\mathbf{w}_0 = r_0\mathbf{e}_1$, where $\mathbf{e}_1$ is the smallest eigenvector direction of Hessian $\mathcal{H} := \nabla^2 F(\mathbf{x}_{m_s})$.

$$
\begin{aligned}
\mathbf{w}_t &= \mathbf{w}_{t-1} - \eta(\mathbf{d}_{t-1} - \mathbf{d}_{t-1}') \\
&= \mathbf{w}_{t-1} - \eta(\nabla F(\mathbf{x}_{t-1}) - \nabla F(\mathbf{x}_{t-1}') + \mathbf{d}_{t-1} - F(\mathbf{x}_{t-1}) - \mathbf{d}_{t-1}' + \nabla F(\mathbf{x}_{t-1}')) \\
&= \mathbf{w}_{t-1} - \eta\Bigg[(\mathbf{x}_{t-1} - \mathbf{x}_{t-1}')\int_0^1 \nabla^2 F(\mathbf{x}_{t-1}' + \theta(\mathbf{x}_{t-1} - \mathbf{x}_{t-1}'))d\theta \\
&\qquad + \mathbf{d}_{t-1} - F(\mathbf{x}_{t-1}) - \mathbf{d}_{t-1}' + F(\mathbf{x}_{t-1}')\Bigg] \\
&= (1 - \eta\mathcal{H})\mathbf{w}_{t-1} - \eta(\Delta_{t-1}\mathbf{w}_{t-1} + \mathbf{y}_{t-1}),
\end{aligned}
$$

where

$$
\begin{aligned}
\Delta_{t-1} &:= \int_0^1 \left(\nabla^2 F(\mathbf{x}_{t-1}' + \theta(\mathbf{x}_{t-1} - \mathbf{x}_{t-1}')) - \mathcal{H}\right)d\theta, \\
\mathbf{y}_{t-1} &:= \mathbf{d}_{t-1} - \nabla F(\mathbf{x}_{t-1}) - \mathbf{d}_{t-1}' + \nabla F(\mathbf{x}_{t-1}') = \boldsymbol{\epsilon}_{t-1} - \boldsymbol{\epsilon}_{t-1}'.
\end{aligned}
$$

Recursively applying the above equation, we get

$$
\mathbf{w}_t = (1 - \eta\mathcal{H})^{t-m_s-1}\mathbf{w}_{m_s+1} - \eta\sum_{\tau=m_s+1}^{t-1}(1 - \eta\mathcal{H})^{t-1-\tau}(\Delta_\tau\mathbf{w}_\tau + \mathbf{y}_\tau).
\tag{C.1}
$$

We want to show that the first term of (C.1) dominates the second term. Next Lemma is essential for the proof of Lemma B.3, which bounds the norm of $\mathbf{y}_t$.

**Lemma C.1.** Under Assumption 3.1, we have following inequality holds,

$$
\begin{aligned}
\|\mathbf{y}_t\|_2 &\leq 2\sqrt{\log(4/\delta)}b^{-1/2}a^{-1/2}\Big(2L\max_{m_s<\tau<t}\|\mathbf{w}_{\tau+1} - \mathbf{w}_\tau\|_2 \\
&\quad + \max_{m_s<\tau\leq t}(2aL + 4\rho D_\tau)\cdot\max_{m_s<\tau\leq t}\|\mathbf{w}_\tau\|_2\Big) + 4\sqrt{\log(4/\delta)}b^{-1/2}Lr_0,
\end{aligned}
\tag{C.2}
$$

where $D_\tau = \max\{\|\mathbf{x}_\tau - \mathbf{x}_{m_s}\|_2, \|\mathbf{x}_\tau' - \mathbf{x}_{m_s}\|_2\}$.

*Proof of Lemma C.1.* By Proposition B.1, we have that

$$
\frac{\mathbf{y}_{t+1}}{(1-a)^{t+1}} - \frac{\mathbf{y}_t}{(1-a)^t} = \frac{\boldsymbol{\epsilon}_{t+1}}{(1-a)^{t+1}} - \frac{\boldsymbol{\epsilon}_t}{(1-a)^t} - \frac{\boldsymbol{\epsilon}_{t+1}'}{(1-a)^{t+1}} + \frac{\boldsymbol{\epsilon}_t'}{(1-a)^t}
$$

$$= \frac{1}{(1-a)^{t+1}} \sum_{i \leq b} [\boldsymbol{\epsilon}_{t,i} - \boldsymbol{\epsilon}'_{t,i}],$$

where $\boldsymbol{\epsilon}_{t,i}$ is the same as that in Proposition B.1:

$$\boldsymbol{\epsilon}_{t,i} = \frac{a}{b} [\nabla f(\mathbf{x}_{t+1}; \boldsymbol{\xi}^i_{t+1}) - \nabla F(\mathbf{x}_{t+1})]$$
$$+ \frac{1-a}{b} [\nabla F(\mathbf{x}_t) - \nabla f(\mathbf{x}_t; \boldsymbol{\xi}^i_{t+1}) - \nabla F(\mathbf{x}_{t+1}) + \nabla f(\mathbf{x}_{t+1}; \boldsymbol{\xi}^i_{t+1})]$$
$$= \frac{1}{b} [\nabla f(\mathbf{x}_{t+1}; \boldsymbol{\xi}^i_{t+1}) - \nabla F(\mathbf{x}_{t+1})] + \frac{1-a}{b} [\nabla F(\mathbf{x}_t) - \nabla f(\mathbf{x}_t; \boldsymbol{\xi}^i_{t+1})], \qquad \text{(C.3)}$$

where we rewrite $\boldsymbol{\epsilon}_{t,i}$ as (C.3) because now we want bound the $\boldsymbol{\epsilon}_t - \boldsymbol{\epsilon}'_t$ by the distance between two sequence. $\boldsymbol{\epsilon}'_{t,i}$ is defined similarly as follows

$$\boldsymbol{\epsilon}'_{t,i} = \frac{1}{b} [\nabla f(\mathbf{x}_{t+1}; \boldsymbol{\xi}^i_{t+1}) - \nabla F(\mathbf{x}_{t+1})] + \frac{1-a}{b} [\nabla F(\mathbf{x}_t) - \nabla f(\mathbf{x}_t; \boldsymbol{\xi}^i_{t+1})].$$

It is easy to verify that $\{\boldsymbol{\epsilon}_{t,i} - \boldsymbol{\epsilon}'_{t,i}\}$ forms a martingale difference sequence. We now bound $\|\boldsymbol{\epsilon}_{t,i} - \boldsymbol{\epsilon}_{t,i'}\|^2_2$. Denote $\mathcal{H}_{t+1,i} = \nabla^2 f(\mathbf{x}_{m_s}; \boldsymbol{\xi}^i_{t+1})$, then we introduce two terms

$$\Delta_{t+1,i} := \int_0^1 (\nabla^2 f(\mathbf{x}'_{t+1} + \theta(\mathbf{x}_{t+1} - \mathbf{x}'_{t+1}); \boldsymbol{\xi}^i_{t+1}) - \mathcal{H}_{t+1,i}) d\theta$$
$$\widehat{\Delta}_{t+1,i} := \int_0^1 (\nabla^2 f(\mathbf{x}'_t + \theta(\mathbf{x}_t - \mathbf{x}'_t); \boldsymbol{\xi}^i_{t+1}) - \mathcal{H}_{t+1,i}) d\theta,$$

By Assumption 3.1, we have $\|\Delta_{t+1,i}\|_2 \leq \rho \max_{\theta \in [0,1]} \|\mathbf{x}'_{t+1} + \theta(\mathbf{x}_{t+1} - \mathbf{x}'_{t+1}) - \mathbf{x}_{m_s+1}\|_2 \leq \rho D_{t+1}$, similarly we have $\|\widehat{\Delta}_{t+1,i}\|_2 \leq \rho D_t$ and $\Delta_{t+1} \leq \rho D_{t+1}$.

Now we bound $\boldsymbol{\epsilon}_{t,i} - \boldsymbol{\epsilon}'_{t,i}$,

$$b(\boldsymbol{\epsilon}_{t,i} - \boldsymbol{\epsilon}'_{t,i}) = \left( [\nabla f(\mathbf{x}_{t+1}; \boldsymbol{\xi}^i_{t+1}) - \nabla F(\mathbf{x}_{t+1})] + (1-a)[\nabla F(\mathbf{x}_t) - \nabla f(\mathbf{x}_t; \boldsymbol{\xi}^i_{t+1})] \right)$$
$$- \left( [\nabla f(\mathbf{x}'_{t+1}; \boldsymbol{\xi}^i_{t+1}) - \nabla F(\mathbf{x}'_{t+1})] - (1-a)[\nabla F(\mathbf{x}'_t) - \nabla f(\mathbf{x}'_t; \boldsymbol{\xi}^i_{t+1})] \right)$$
$$= (\mathcal{H}_{t+1,i}\mathbf{w}_{t+1} + \Delta_{t+1,i}\mathbf{w}_{t+1} - \mathcal{H}\mathbf{w}_{t+1} - \Delta_{t+1}\mathbf{w}_{t+1} + (1-a)\mathcal{H}\mathbf{w}_t$$
$$+ (1-a)\Delta_t\mathbf{w}_t - (1-a)\mathcal{H}_{t+1,i}\mathbf{w}_t - (1-a)\widehat{\Delta}_{t+1,i}\mathbf{w}_t)$$
$$= (\mathcal{H}_{t+1,i} - \mathcal{H})(\mathbf{w}_{t+1} - (1-a)\mathbf{w}_t) + (\Delta_{t+1,i} - \Delta_{t+1})\mathbf{w}_{t+1}$$
$$+ (1-a)(\Delta_t - \widehat{\Delta}_{t+1,i})\mathbf{w}_t. \qquad \text{(C.4)}$$

This implies the LHS of (C.4) has the following bound.

$$\|b(\boldsymbol{\epsilon}_{t,i} - \boldsymbol{\epsilon}'_{t,i})\|_2 \leq 2L\|\mathbf{w}_{t+1} - (1-a)\mathbf{w}_t\|_2 + 2\rho D^x_{t+1}\|\mathbf{w}_{t+1}\|_2 + 2\rho D^x_t\|\mathbf{w}_t\|_2$$
$$\leq 2L\|\mathbf{w}_{t+1} - \mathbf{w}_t\|_2 + 2\rho D^x_{t+1}\|\mathbf{w}_{t+1}\|_2 + (2aL + 2\rho D^x_t)\|\mathbf{w}_t\|_2$$
$$\leq \underbrace{2L \max_{m_s < \tau < t} \|\mathbf{w}_{\tau+1} - \mathbf{w}_\tau\|_2 + \max_{m_s < \tau \leq t} (2aL + 4\rho D_\tau) \cdot \max_{m_s < \tau \leq t} \|\mathbf{w}_\tau\|_2}_{M}$$

where the first inequality is by the gradient Lipschitz Assumption and Hessian Lipschitz Assumption 3.1, the second inequality is by triangle inequality. Therefore we have

$$\|\boldsymbol{\epsilon}_{t,i} - \boldsymbol{\epsilon}'_{t,i}\|^2_2 \leq \frac{M^2}{b^2}$$

Furthermore, by Azuma Hoeffding inequality(See Lemma D.1 for detail), with probability at least $1 - \delta$, we have that for any $t > 0$,

$$\left\| \frac{\mathbf{y}_t}{(1-a)^t} - \frac{\mathbf{y}_{m_s+1}}{(1-a)^{m_s+1}} \right\|^2_2 = \left\| \sum_{\tau=m_s+1}^{t-1} \left( \frac{\mathbf{y}_{\tau+1}}{(1-a)^{\tau+1}} - \frac{\mathbf{y}_\tau}{(1-a)^\tau} \right) \right\|^2_2$$

$$= \left\| \sum_{\tau=m_s+1}^{t-1} \left( \frac{1}{(1-a)^{\tau+1}} \sum_{i \leq b} [\boldsymbol{\epsilon}_{\tau,i} - \boldsymbol{\epsilon}'_{\tau,i}] \right) \right\|_2^2$$

$$\leq 4\log(4/\delta) \left( \sum_{i=m_s+1}^{t-1} b \cdot \frac{M^2}{(1-a)^{2\tau+2}b^2} \right).$$

Multiply $(1-a)^{2t}$ on both side, we get

$$\|\mathbf{y}_t - (1-a)^{t-m_s-1}\mathbf{y}_{m_s+1}\|_2^2 \leq 4b^{-1}\log(4/\delta) \sum_{\tau=m_s+1}^{t-1} (1-a)^{2t-2\tau-2}M^2$$

$$\leq 4\log(4/\delta)b^{-1}a^{-1}M^2,$$

where the last inequality is by $\sum_{i=0}^{t-1}(1-a)^{2t-2i-2} \leq a^{-1}$. Furthermore, by triangle inequality we have

$$\|\mathbf{y}_t\|_2 \leq 2\sqrt{log(4/\delta)}b^{-1/2}a^{-1/2}M + (1-a)^{t-m_s-1}\|\mathbf{y}_{m_s+1}\|_2. \tag{C.5}$$

$\|\nabla f(\mathbf{x}_{m_s+1}; \boldsymbol{\xi}_{m_s+1}^i) - \nabla F(\mathbf{x}'_{m_s+1}) - \nabla f(\mathbf{x}'_{m_s+1}; \boldsymbol{\xi}_{m_s+1}^i) + \nabla F(\mathbf{x}'_{m_s+1})\|_2 \leq 2Lr_0$ due to Assumption 3.1. Then by Azuma Inequality(See Lemma D.1), we have with probability at least $1 - \delta$,

$$\|\mathbf{y}_{m_s+1}\|_2^2 = \|\mathbf{d}_{m_s+1} - \nabla F(\mathbf{x}_{m_s+1}) - \mathbf{d}'_{m_s+1} + \nabla F(\mathbf{x}'_{m_s+1})\|_2^2$$

$$= \left\| \frac{1}{b} \sum_{i \leq b} [\nabla f(\mathbf{x}_{m_s+1}; \boldsymbol{\xi}_{m_s+1}^i) - \nabla F(\mathbf{x}'_{m_s+1}) - \nabla f(\mathbf{x}'_{m_s+1}; \boldsymbol{\xi}_{m_s+1}^i) + \nabla F(\mathbf{x}'_{m_s+1})] \right\|_2^2$$

$$\leq \frac{4\log(4/\delta)4L^2r_0^2}{b}. \tag{C.6}$$

Plugging (C.6) into (C.5) gives

$$\|\mathbf{y}_t\|_2 \leq 2\sqrt{\log(4/\delta)}b^{-1/2}a^{-1/2}\Big(2L \max_{m_s < \tau < t} \|\mathbf{w}_{\tau+1} - \mathbf{w}_\tau\|_2$$

$$+ \max_{m_s < \tau \leq t}(2aL + 4\rho D_\tau) \cdot \max_{m_s < \tau \leq t} \|\mathbf{w}_\tau\|_2\Big) + 4\sqrt{\log(4/\delta)}b^{-1/2}Lr_0.$$

$\square$

Now we can give a proof of Lemma B.3.

*Proof of Lemma B.3.* We proof it by induction that

1. $\frac{1}{2}(1 + \eta_H\gamma)^{t-m_s-1}r_0 \leq \|\mathbf{w}_t\|_2 \leq \frac{3}{2}(1 + \eta_H\gamma)^{t-m_s-1}r_0$.

2. $\|y_t\|_2 \leq 2\eta_H\gamma L(1 + \eta_H\gamma)^{t-m_s-1}r_0$.

First for $t = m_s + 1$, we have $\|\mathbf{w}_{m_s+1}\|_2 = r_0$, $\|y_{m_s+1}\|_2 \leq \sqrt{16b^{-1}\log(4/\delta)L^2r_0^2} \leq 2\eta_H\gamma Lr_0$(See (C.6)), where $b \geq 2\eta_H^{-2}\gamma^{-2}\sqrt{\log(4/\delta)}$. Assume they hold for all $m_s < \tau < t$, we now prove they hold for t. We bound $\mathbf{w}_t$ first, we only need to show that second term of (C.1) is bounded by $\frac{1}{2}(1 + \eta_H\gamma)^t r_0$.

$$\left\| \eta_H \sum_{\tau=m_s+1}^{t-1} (1 - \eta_H\mathcal{H})^{t-1-\tau}(\Delta_\tau\mathbf{w}_\tau + \mathbf{y}_\tau) \right\|_2$$

$$\leq \eta_H \sum_{\tau=m_s+1}^{t-1} (1 + \eta_H\gamma)^{t-1-\tau}(\|\Delta_\tau\|_2\|\mathbf{w}_\tau\|_2 + \|\mathbf{y}_\tau\|_2)$$

$$\leq \eta_H \sum_{\tau=m_s+1}^{t-1} (1+\eta_H\gamma)^{t-m_s-2} r_0 \left(\frac{3}{2}\|\Delta_\tau\|_2 + 2\eta_H\gamma L\right)$$

$$\leq \eta_H \sum_{\tau=m_s+1}^{t-1} (1+\eta_H\gamma)^{t-m_s-2} r_0 (3\eta_H\epsilon_H L + 2\eta_H\gamma L)$$

$$= \eta_H \ell (1+\eta_H\gamma)^{t-m_s-2} r_0 \cdot 5\eta_H\gamma L$$

$$\leq 10\log(8\epsilon_H\rho^{-1}r_0^{-1})\eta_H L(1+\eta_H\gamma)^{t-m_s-2} r_0$$

$$\leq \frac{1}{2}(1+\eta_H\gamma)^{t-m_s-1} r_0,$$

where the first inequality is by the eigenvalue assumption over $\mathcal{H}$, the second inequality is by the Induction hypothesis, the third inequality is by $\|\Delta_\tau\|_2 \leq \rho D_\tau = \rho\max\{\|\mathbf{x}_\tau - \mathbf{x}_{m_s}\|_2, \|\mathbf{x}'_\tau - \mathbf{x}_{m_s}\|_2\} \leq \eta_H\epsilon_H L + r\rho \leq 2\eta_H\epsilon_H L$, the fourth inequality is by the choice of $t - m_s - 1 \leq \ell \leq 2\log(8\epsilon_H\rho^{-1}r_0^{-1})/(\eta_H\gamma)$, the last inequality is by the choice of $\eta_H \leq 1/(10\log(8\epsilon_H\rho^{-1}r_0^{-1})L)$. Now we bound $\|\mathbf{y}_t\|_2$ by (C.2). We first get the bound for $L\|\mathbf{w}_{i+1} - \mathbf{w}_i\|_2$ as follows,

$$L\|\mathbf{w}_{t+1} - \mathbf{w}_t\|_2$$

$$= L\left\| -\eta_H\mathcal{H}(I-\eta_H\mathcal{H})^{t-m_s-2}\mathbf{w}_0 - \eta_H\sum_{\tau=m_s+1}^{t-2}\eta_H\mathcal{H}(I-\eta_H\mathcal{H})^{t-2-\tau}(\Delta_\tau\mathbf{w}_\tau + \mathbf{y}_\tau) \right.$$

$$\left. + \eta_H(\Delta_{t-1}\mathbf{w}_{t-1} + \mathbf{y}_{t-1}) \right\|_2$$

$$\overset{(i)}{\leq} L\eta_H\gamma(1+\eta_H\gamma)^{t-m_s-2}r_0 + L\eta_H\left\| \sum_{\tau=m_s+1}^{t-2}\eta_H\mathcal{H}(I-\eta_H\mathcal{H})^{t-2-\tau}(\Delta_\tau\mathbf{w}_\tau + \mathbf{y}_\tau) \right\|_2$$

$$+ L\eta_H\left\| \Delta_{t-1}\mathbf{w}_{t-1} + \mathbf{y}_{t-1} \right\|_2$$

$$\overset{(ii)}{\leq} L\eta_H\gamma(1+\eta_H\gamma)^{t-m_s-2}r_0$$

$$+ L\eta_H\left[\left\| \sum_{\tau=m_s+1}^{t-2}\eta_H\mathcal{H}(I-\eta_H\mathcal{H})^{t-2-\tau} \right\|_2 + 1\right]\max_{0\leq\tau\leq t-1}\left\| \Delta_\tau\mathbf{w}_\tau + \mathbf{y}_\tau \right\|_2$$

$$\overset{(iii)}{\leq} L\eta_H\gamma(1+\eta_H\gamma)^{t-m_s-2}r_0 + L\eta_H\left[\sum_{\tau=m_s+1}^{t-2}\frac{1}{t-1-\tau} + 1\right]\max_{0\leq\tau\leq t-1}\left\| \Delta_\tau\mathbf{w}_\tau + \mathbf{y}_\tau \right\|_2$$

$$\overset{(iv)}{\leq} L\eta_H\gamma(1+\eta_H\gamma)^{t-m_s-2}r_0 + L\eta_H[\log(t-m_s-1)+1]\cdot[5\eta_H\gamma L(1+\eta_H\gamma)^{t-m_s-2}r_0]$$

$$\overset{(v)}{\leq} 6L\eta_H\gamma(1+\eta_H\gamma)^{t-m_s-2}r_0 + 5\log(t-m_s-1)\gamma\eta_H^2 L^2(1+\eta_H\gamma)^{t-m_s-2}r_0, \tag{C.7}$$

where (i) is by triangle inequality, (ii) is by the definition of max, (iii) is by $\|\eta_H\mathcal{H}(I-\eta_H\mathcal{H})^{t-2-\tau}\|_2 \leq \frac{1}{t-1-\tau}$, (iv) is due to $\|\Delta_\tau\|_2 \leq \rho D_\tau \leq \rho(\eta_H\gamma L/\rho + r) \leq 2\gamma\eta_H L$, $\|\mathbf{w}_\tau\|_2 \leq 3(1+\eta_H\gamma)^{\tau-m_s-1}r_0/2$ and $\|\mathbf{y}_\tau\|_2 \leq 2\eta_H\gamma L(1+\eta_H\gamma)^{\tau-m_s-1}r_0$, (v) is due to $\eta_H \leq 1/L$. We next get the bound of $\max_{m_s<\tau\leq t}(2aL + 4\rho D_\tau)\cdot\max_{m_s<\tau\leq t}\|\mathbf{w}_\tau\|_2$ as follows

$$\max_{m_s<\tau\leq t}(2aL + 4\rho D_\tau)\cdot\max_{m_s<\tau\leq t}\|\mathbf{w}_\tau\|_2 \leq (2aL + 8\gamma\eta_H L)\frac{3(1+\eta_H\gamma)^{t-m_s-1}}{2}r_0$$

$$\leq 15\gamma\eta_H L(1+\eta_H\gamma)^{t-m_s-1}r_0. \tag{C.8}$$

where the first inequality is by $\rho D_t \leq \rho(\gamma\eta_H L/\rho + r) \leq 2\gamma\eta_H L$ and the induction hypothesis, last inequality is by $a \leq \gamma\eta_H$.

Plugging (C.7) and (C.8) into (C.2) gives,

$$\|\mathbf{y}_t\|_2 \leq 2\sqrt{\log(4/\delta)}b^{-1/2}a^{-1/2}\left(2L\max_{m_s<\tau<t}\|\mathbf{w}_{\tau+1} - \mathbf{w}_\tau\|_2\right.$$

$$+ \max_{m_s < \tau \leq t} (2aL + 4\rho D_\tau) \cdot \max_{m_s < \tau \leq t} \|\mathbf{w}_\tau\|_2 \Big) + 4\sqrt{\log(4/\delta)}b^{-1/2}Lr_0$$

$$\leq 2\sqrt{\log(4/\delta)}b^{-1/2}a^{-1/2}\Big(10\log(\ell)\gamma\eta_H^2 L^2(1+\eta_H\gamma)^{t-m_s-1}r_0$$

$$+ 27\gamma\eta_H L(1+\eta_H\gamma)^{t-m_s-1}r_0\Big) + 4\sqrt{\log(4/\delta)}b^{-1/2}Lr_0$$

$$\leq \underbrace{56\sqrt{\log(4/\delta)}b^{-1/2}a^{-1/2}\eta_H L\gamma(1+\eta_H\gamma)^{t-m_s-1}r_0}_{I_1}$$

$$+ \underbrace{4\sqrt{\log(4/\delta)}b^{-1/2}(1+\eta_H\gamma)^{t-m_s-1}r_0}_{I_2}$$

where the last inequality is by $\eta_H \leq 1/(10L\log\ell)$. Now we bound $I_1$ and $I_2$ respectively.

$$I_1 = 56\sqrt{\log(4/\delta)}b^{-1/2}a^{-1/2}\eta_H L\gamma(1+\eta_H\gamma)^{t-m_s-1}r_0$$

$$= \eta_H\gamma L(1+\eta_H\gamma)^{t-m_s-1}r_0,$$

where the inequality is applying $b \geq 56^2\log(4/\delta)a^{-1}$. Now we bound $I_2$ by applying $b \geq 16\log(4/\delta)\eta_H^{-2}L^{-2}\gamma^{-2}$,

$$I_2 \leq \eta_H\gamma L(1+\eta_H\gamma)^{t-m_s-1}r_0.$$

Then we obtain that

$$\|\mathbf{y}_t\|_2 \leq 2\eta_H\gamma L(1+\eta_H\gamma)^{t-m_s-1}r_0,$$

which finishes the induction. So we have $\|\mathbf{w}_t\|_2 \geq \frac{1}{2}(1+\eta_H\gamma)^{t-m_s-1}r_0$. However, the triangle inequality give the bound

$$\|\mathbf{w}_t\|_2 \leq \|\mathbf{x}_t - \mathbf{x}_{m_{s+1}}\|_2 + \|\mathbf{x}_{m_{s+1}} - \mathbf{x}_{m_s}\|_2 + \|\mathbf{x}_t' - \mathbf{x}_{m_{s+1}}'\|_2 + \|\mathbf{x}_{m_s+1}' - \mathbf{x}_{m_s}'\|_2$$

$$\leq 2r + 2\frac{\epsilon_H\eta_H L}{\rho}$$

$$\leq 4\frac{\epsilon_H\eta_H L}{\rho},$$

where the last inequality is due to $r \leq \epsilon_H\eta_H L/\rho$. So we obtain that

$$t \leq \frac{\log(8\epsilon_H\eta_H L\rho^{-1}r_0^{-1})}{\log(1+\eta_H\gamma)} < \frac{2\log(8\epsilon_H\rho^{-1}r_0^{-1})}{\eta_H\gamma}.$$

$\square$

# D   Auxiliary Lemmas

We start by providing the Azuma–Hoeffding inequality under the vector settings.

**Lemma D.1** (Theorem 3.5, [24]). Let $\boldsymbol{\epsilon}_{1:k} \in \mathbb{R}^d$ be a vector-valued martingale difference sequence with respect to $\mathcal{F}_k$, i.e., for each $k \in [K]$, $\mathbb{E}[\boldsymbol{\epsilon}_k|\mathcal{F}_k] = 0$ and $\|\boldsymbol{\epsilon}_k\|_2 \leq B_k$, then we have given $\delta \in (0,1)$, w.p. $1-\delta$,

$$\left\|\sum_{i=1}^{K}\boldsymbol{\epsilon}_k\right\|_2^2 \leq 4\log(4/\delta)\sum_{i=1}^{K}B_k^2.$$

This lemma provides a dimension-free bound due to the fact that the Euclidean norm version of $\mathbb{R}^d$ is $(2,1)$ smooth, see also [15, 9]. Now, we are give a proof of Lemma 6.1.

We have the following lemma:

**Lemma D.2.** For any $t \neq m_s$, we have

$$F(\mathbf{x}_{t+1}) \leq F(\mathbf{x}_t) - \frac{\eta_t}{2}\|\mathbf{d}_t\|_2^2 + \frac{\eta_t}{2}\|\boldsymbol{\epsilon}_t\|_2^2 + \frac{L}{2}\|\mathbf{x}_{t+1} - \mathbf{x}_t\|_2^2.$$

For $t = m_s$, we have $F(\mathbf{x}_{t+1}) \leq F(\mathbf{x}_t) + (\|\mathbf{d}_t\|_2 + \|\boldsymbol{\epsilon}_t\|_2 + Lr/2)r$.

586 *Proof of Lemma D.2.* By Assumption 3.1, we have

$$F(\mathbf{x}_{t+1}) \le F(\mathbf{x}_t) + \langle \nabla F(\mathbf{x}_t), \mathbf{x}_{t+1} - \mathbf{x}_t \rangle + \frac{L}{2}\|\mathbf{x}_{t+1} - \mathbf{x}_t\|_2^2. \tag{D.1}$$

587 For the case $t \ne m_s$, the update rule is $\mathbf{x}_{t+1} = \mathbf{x}_t - \eta_t \mathbf{d}_t$, therefore

$$
\begin{aligned}
F(\mathbf{x}_{t+1}) &\le F(\mathbf{x}_t) - \eta_t \langle \nabla F(\mathbf{x}_t), \mathbf{d}_t \rangle + \frac{L}{2}\|\mathbf{x}_{t+1} - \mathbf{x}_t\|_2^2 \\
&= F(\mathbf{x}_t) - \eta_t \|\nabla F(\mathbf{x}_t)\|_2^2/2 - \eta_t\|\mathbf{d}_t\|_2^2/2 + \eta_t\|\boldsymbol{\epsilon}_t\|_2^2/2 + L\|\mathbf{x}_{t+1} - \mathbf{x}_t\|_2^2/2 \\
&\le F(\mathbf{x}_t) - \eta_t \|\mathbf{d}_t\|_2^2/2 + \eta_t\|\boldsymbol{\epsilon}_t\|_2^2/2 + \frac{L}{2}\|\mathbf{x}_{t+1} - \mathbf{x}_t\|_2^2,
\end{aligned}
$$

588 where the first inequality on the first line is due to Assumption 3.1 and the second inequality holds
589 trivially. For the case $t = m_s$, since $\|\nabla F(\mathbf{x}_t)\|_2 \le \|\mathbf{d}_t\|_2 + \|\boldsymbol{\epsilon}_t\|_2$ we have

$$
\begin{aligned}
F(\mathbf{x}_{t+1}) &\le F(\mathbf{x}_t) + \langle \nabla F(\mathbf{x}_t), \mathbf{x}_{t+1} - \mathbf{x}_t \rangle + \frac{L}{2}\|\mathbf{x}_{t+1} - \mathbf{x}_t\|_2^2 \\
&\le F(\mathbf{x}_t) + (\|\mathbf{d}_t\|_2 + \|\boldsymbol{\epsilon}_t\|_2 + Lr/2)r.
\end{aligned}
$$

590 $\qquad\qquad\qquad\qquad\qquad\qquad\qquad\qquad\qquad\qquad\qquad\qquad\qquad\qquad\qquad\qquad\qquad\qquad\quad\square$

591 **Lemma D.3** (Lemma 6, [17])**.** Suppose $-\gamma = \lambda_{\min}(\nabla^2 F(\mathbf{x}_{m_s})) \le -\epsilon_H$. Set $r \le$
592 $L\eta_H\epsilon_H/(C\rho)$, $\ell_{\text{thres}} = 2\log(\eta_H\epsilon_H\sqrt{d}LC^{-1}\rho^{-1}\delta^{-1}r^{-1})/(\eta_H\epsilon_H) = \widetilde{O}(\eta_H^{-1}\epsilon_H^{-1})$, $\eta_H \le$
593 $\min\{1/(16L\log(\eta_H\epsilon_H\sqrt{d}LC^{-1}\rho^{-1}\delta^{-1}r^{-1})), 1/(8CL\log\ell_{\text{thres}})\} = \widetilde{O}(L^{-1})$, $b = q = \sqrt{B} \ge$
594 $16\log(4/\delta)/(\eta_H^2\epsilon_H^2)$. Let $\{\mathbf{x}_t\}, \{\mathbf{x}_t'\}$ be two coupled sequences by running `Pullback-SPIDER`
595 from $\mathbf{x}_{m_s+1}, \mathbf{x}_{m_s+1}'$ with $\mathbf{w}_{m_s+1} = \mathbf{x}_{m_s+1} - \mathbf{x}_{m_s+1}' = r_0\mathbf{e}_1$, where $\mathbf{x}_{m_s+1}, \mathbf{x}_{m_s+1}' \in \mathbb{B}_{\mathbf{x}_{m_s}}(r)$,
596 $r_0 = \delta r/\sqrt{d}$ and $\mathbf{e}_1$ denotes the smallest eigenvector direction of Hessian $\nabla^2 F(\mathbf{x}_{m_s})$. Then with
597 probability at least $1 - \delta$,

$$\max_{m_s < t < m_s + \ell_{\text{thres}}} \{\|\mathbf{x}_t - \mathbf{x}_{m_s}\|_2, \|\mathbf{x}_0 - \mathbf{x}_{m_s}\|_2\} \ge \frac{L\eta_H\epsilon_H}{C\rho}, \tag{D.2}$$

598 where $C = O(\log(d\ell_{\text{thres}}/\delta)) = \widetilde{O}(1)$.