# OpenReview forum: "Faster Perturbed Stochastic Gradient Methods for Finding Local Minima"
_NeurIPS.cc/2021/Conference — NeurIPS 2021 Submitted_

### Official Review · Reviewer_zSqf · 2021-06-28

**Rating:** 5
**Confidence:** 3

**Summary:**

The paper proposes a new approach called "Pullback" for escaping saddle points and finding local minimum that matches the same complexity as SGD. The approach perturbs a stochastic gradient estimators (SARAH/SPIDER) and STORM by an element uniformly on the ball of radius $r$. The proposed method is able to find an $(\varepsilon, \sqrt{\varepsilon})$-approximate local minima within $\mathcal{O}(\varepsilon^{-3.5})$ stochastic gradient evaluations. This is the fastest known algorithm achieving the optimal rate of $\mathcal{O}(\varepsilon^{-3})$. The proof techniques appear to be similar to previous works and the gradient estimator used to achieve this result is known.

**Limitations And Societal Impact:**

The authors have addressed the limitations and potential negative societal impact their work. This work is purely theoretically and mainly combining known algorithms.

**Main Review:**

The paper is well-written and the introduction provides a sufficient background to understand the question of interest: "Can perturbed SGD methods find local minima efficiently?" Most of the paper is devoted to explaining the algorithm, which is done quite clearly, and a sketch of the proof. While I did not check all the details of the proof, the proof does appear correct and I appreciated that all the assumptions were clearly stated.

While I appreciate the theoretical question of finding fastest known algorithm for escaping saddle points under the stochastic setting, I would like the author to elucidate why this question is of broader importance to the ML community. While I understand there is a line of research in this area, it is not clear to me that saddle points are even a problem when one uses a stochastic algorithm. Would the author be able to point to literature which shows that saddle points appear in large-scale problems and in particular, stochastic algorithms would get stuck in them? Or more broadly why finding a local minima in the stochastic setting is hard? I would highly suggest that the author add some experiments (simulated and/or real data) which (1). shows that the proposed algorithm is implementable and (2). if possible, generate a problem that has saddle points and run the proposed algorithm on such a problem.

**Time Spent Reviewing:**

3

---

> ### Author Response · Authors · 2021-08-10
> **Response to Reviewer zSqf**
>
> Thanks for your detailed comments!
>
> -----
>
> **Q1:** The proof techniques appear to be similar to previous works.
>
> **A1:** Our proof technique is actually quite different from previous works. Existing algorithms [arxiv.1807.01695] which can achieve the $\tilde O(\epsilon^{-3})$ complexity is based on the negative curvature search subroutine. In contrast, our proposed algorithm is based on the perturbed stochastic gradients and enjoys the $\tilde O(\epsilon^{-3})$ complexity, whose proof strategy and technique are quite different from those of the negative curvature-based algorithms. Existing algorithms based on perturbed-gradient can only achieve $\tilde{O}(poly(\epsilon^{-1}))$ [arxiv.1503.02101], $\tilde O(\epsilon^{-5})$ [arxiv.1803.05999] or $\tilde O(\epsilon^{-3.5})$ [arxiv.1904.09265] gradient complexity. In order to obtain faster convergence rates, we have made a nontrivial effort in both algorithm design and analysis.
>
> ---
> **Q2:**  "While I appreciate the theoretical question of finding fastest known algorithm for escaping saddle points under the stochastic setting, I would like the author to elucidate why this question is of broader importance to the ML community. While I understand there is a line of research in this area, it is not clear to me that saddle points are even a problem when one uses a stochastic algorithm."
>
> **A2:** We believe escaping saddle points is of great importance to the ML community. For example, many problems in ML such as semi-definite programming, matrix sensing and matrix completion can be converted into nonconvex optimization problems and solved by (stochastic) gradient descent. For this class of nonconvex problems, they enjoy a particular property that all local minima are global minima. Therefore, escaping from saddle points faster immediately implies finding global minima faster, which is indeed of great importance to the ML community who are interested in these problems. In fact, escaping saddle points and finding approximate local minima is one of the central research questions in nonconvex optimization. If you look at the added experiments on the matrix sensing problem (please see "Added Experiments" at the beginning), you can see that both SGD and SPIDER cannot escape the saddle points. This indicates that for certain problems, saddle points are still a problem even when one uses a stochastic algorithm. We will add more discussion to motivate this research question in the revision.
>
> ---
>
> **Q3:**  "Would the author be able to point to literature which shows that saddle points appear in large-scale problems and in particular, stochastic algorithms would get stuck in them? "
>
> **A3:** For example, [arxiv.1406.2572] has shown that the main bottleneck in the optimization of neural networks is not due to local minima but the existence of many saddle points. Thus we believe escaping from saddle points is an important question even in deep learning.
>
> ---
>
> **Q4:** "Or more broadly why finding a local minima in the stochastic setting is hard? I would highly suggest that the author add some experiments (simulated and/or real data) which (1). shows that the proposed algorithm is implementable and (2). if possible, generate a problem that has saddle points and run the proposed algorithm on such a problem".
>
> **A4:** Thank you for your suggestions. We have performed experiments on the nonconvex matrix sending problem. Please see 'Added Experiments' at the beginning for more details. From the table, we can see that SGD and SARAH/SPIDER were stuck at saddle point with objection value around 0.6 while using perturbation or second-order information can effectively escape saddle points.

---

### Official Review · Reviewer_AFDY · 2021-07-17

**Rating:** 6
**Confidence:** 4

**Summary:**

This papers looks into stochastic gradient methods with the goal of avoiding saddle points in nonconvex optimization. The main contribution of this work is a *perturbed* stochastic gradient framework (called "Pullback") which, by using a particular normalization scheme, finds an $(\epsilon, \epsilon_H)$-approximate local minimum (that is, an $x$ s.t. $||\nabla F(x)|| \leq \epsilon$ and $\lambda_{\textrm{min}}(\nabla^2 F(x)) \geq -\epsilon_H$) in $\tilde{O}(\epsilon^{-3} + \epsilon_H^{-6})$ stochastic gradient evaluations. When $\epsilon_H = \sqrt{\epsilon}$ (referred to as the "classic setting" in the paper), their method results in a total of $\tilde{O}(\epsilon^{-3})$ stochastic gradient evaluations, which improves upon all previous perturbed methods. This rate also matches (in the classic setting) the rate previously attained by SPIDER-SFO+ (+Neon2) (though this previous method requires an additional negative curvature search routine).

**Limitations And Societal Impact:**

The authors study the theoretical aspect of optimization algorithms, which they have adequately addressed.

**Main Review:**

________  Major Comments  ________

While it is written decently well, there is a key discussion missing from the paper that, in my opinion, is crucial to better placing the work in the right context. To elaborate, the main claim of novelty in the paper is that (focusing on the "classic setting" for the moment) it achieves a $\tilde{O}(\epsilon^{-3})$ rate for reaching an approximate local minimum using a *perturbed* SGD-type method (and so, crucially, *without* needing to rely on a (black-box) negative curvature search routine). This distinction is made because (as mentioned in the summary above), this rate has previously been achieved through a combination of SPIDER-SFO+ with Neon2. This type of simplification---particularly in the absence of improved convergence rates---certainly deserves a justification that goes beyond what is now found in the paper ("In contrast to this line of works, our algorithm is simpler since it does not need to use the negative curvature search routine."). For example, exactly how much more costly is it to run the negative curvature search routine? What additional operations would it entail? Is there a (perhaps empirically motivated) setting in which perturbation really does help?

That said, I do feel this sort of improved understanding is important, in part due to earlier work in finding approximate local minima via perturbed methods (e.g., Ge et al. (2015)), and more generally as a means of validating these (intuitively appealing) perturbation schemes. These methods have also been important in the deterministic setting, e.g. Jin et al. (2017) (as the paper cites), and see also Jin et al. (2018) [3]. Admittedly this work still does not match previous methods in the general $\epsilon_H$ setting (more on that below), but its contributions are a step in the right direction.


________  Minor Comments  ________

- Title: Should be "Finding Approximate Local Minima", rather than "Finding Local Minima"


- p.1: "which matches the number of gradient evaluations for gradient descent to find $\epsilon$-stationary points"

	- It should be clarified that this is true when only smoothness (i.e., Lipschitz continuous gradient) is assumed. With higher-order smoothness (e.g., Lipschitz continuous Hessian), better gradient evaluation complexities are possible. In fact, it would generally be helpful to mention earlier on in the paper what smoothness assumptions are made.


- p.1: "perturbed Stochastic gradient" => "perturbed stochastic gradient"


- p.1: The paper mentions "$\epsilon$-stationary points" without first defining it or contrasting it with approximate local minima.


- p.2: "Besides, we show" => "{Furthermore/In addition/etc.}, we show"


- p.2: "To compare with, we summarized related results" => "To compare with previous methods, we summarize related results"


- p.2: "Comparison of of different optimization algorithm for find approximate local minima of non convex online problems." => "Comparison of different optimization algorithms for finding approximate local minima of nonconvex stochastic problems."


- p.2: "In specific" -> "Specifically"


- p.3: "Assumption 3.2 is standard in online/stochastic optimization for finding second-order stationary points"

	- It would be better to stick to either "approximate local minima" (as defined in the introduction), or "second-order stationary point" (though this is only later defined).

	- In fact, the assumption is standard for settings outside of searching for approximate local minima; perhaps a better phrasing would be: "Assumption 3.2 is standard in online/stochastic optimization, including for finding approximate local minima, and immediately..."


- p.4: "which is in the same order as GD for finding $\epsilon$-stationary points."

	- Again, it depends on the level of smoothness assumed, and so this should be clarified.

"in the same order" => "of the same order"


- p.4: "competative" => "competitive"


- p.4: "big batch of stochastic gradient" => "big batch of stochastic gradients"


- p.4: "is that, it" => "is that it"


- p.5: In Algorithm 1, it would be nicer to e.g. have {"PullBack"} as a (right-aligned) comment (or something similar).


- p.5: "which starts at m_s-th step" => "which starts at the m_s-th step"


- p.5: "well-control" => "carefully control"


- p.6: Regarding the reference in Remark 5.2 to the lower bound of Arjevani et al. (2020): as stated, the remark is misleading, as the lower bound is for stochastic first- and second-order methods. (In contrast, SPIDER and SPIDER-based methods rely on multi-point queries). Arjevani et al. also give a stochastic first- and second-order method (Algorithm 5) which achieves (ignoring Lipschitz and variance parameters) a rate of $\tilde{O}(\epsilon^{-3} + \epsilon^{-2}\epsilon_H^{-2}+\epsilon_H^{-5})$, though it also relies on a negative curvature estimation.


- p.9: "Our results show that simple perturbed gradient methods can be as efficient as more sophisticated algorithms for finding local minima."

	- The authors should clarify here that they are as efficient *in the classic setting*, and are only nearly as efficient in the more general setting.


- It would be useful to mention the following works, given the way they also highlight, in the deterministic setting, the distinction between pertubation-based and non-perturbation-based methods for finding approximate local minima:
	- Agarwal et al. (2017) [1]
	- Carmon et al. (2018) [2]
	- Jin et al. (2018) [3]


[1] Naman Agarwal, Zeyuan Allen-Zhu, Brian Bullins, Elad Hazan, and Tengyu Ma. "Finding approximate local minima faster than gradient descent." In Proceedings of the 49th Annual ACM SIGACT Symposium on Theory of Computing, pp. 1195-1199. 2017.

[2] Yair Carmon, John C. Duchi, Oliver Hinder, and Aaron Sidford. "Accelerated methods for nonconvex optimization." SIAM Journal on Optimization 28, no. 2 (2018): 1751-1772.

[3] Chi Jin, Praneeth Netrapalli, and Michael I. Jordan. "Accelerated gradient descent escapes saddle points faster than gradient descent." In Conference On Learning Theory, pp. 1042-1085. PMLR, 2018.

**Time Spent Reviewing:**

5

---

> ### Author Response · Authors · 2021-08-10
> **Response to Reviewer AFDY**
>
> Thanks for your helpful comments!
>
> ----
> **Q1:** "This type of simplification ... certainly deserves a justification that goes beyond what is now found in the paper ... Is there a (perhaps empirically motivated) setting in which perturbation really does help?"
>
> **A1:** Our algorithm is empirically better than previous NEON2-based algorithms, which can be seen through the experiment results (Please refer to 'Added Experiments' at the beginning). The reasons are as follows. First, the accuracy of the negative curvature estimation is very crucial to the success of NEON2-based algorithms. However, we found that the accuracy heavily depends on the number of iterations in the NEON2 algorithm, which requires careful parameter tuning to balance the computational cost and the accuracy. In contrast, our algorithm only relies on gradient descent-type updates besides an added noise, which is easier to tune. Second, to make NEON2-based algorithms work, a random flip of the output curvature direction is needed (for instance, line 5 in Algorithm 2, [arxiv.1807.01695]). However, in experiments, we find such a flip makes the algorithm unstable, which hurts the final performance. In contrast, our perturbation-based algorithm does not have such an instability issue. We will add more comments on the advantage of perturbation-based algorithms over Neon2-based algorithms in the revision to better motivate our work.
>
> ---
>
> **Q2:**  Distinction between pertubation-based and non-perturbation-based methods for finding approximate local minima [1-3]
>
> **A2:** Thanks for your pointing out these related works. [1-3] focus on finding approximate local minima in the deterministic setting. [1] approximates the cubic regularization method with gradient and Hessian-vector product-based methods. [2] proposes a regularization-based framework that combines accelerated gradient descent and the negative curvature searching method. [3] proposes a perturbed accelerated gradient descent method. All of them can find $(\epsilon, \sqrt{\epsilon})$-approximate local minima within $\tilde O(\epsilon^{-7/4})$ number of gradient/Hessian-vector product evaluations. We will comment on these works in the revision.
>
> ---
> **Q3:** Typos.
>
> **A3:** Thank you for your suggestions, we will fix them.

---

### Official Review · Reviewer_Au4C · 2021-07-17

**Rating:** 7
**Confidence:** 1

**Summary:**

This paper studies the perturbed gradient methods for stochastic optimization to escap from saddle points and find local minimum. New complexity is proved, which is sharper than the previous results.

**Limitations And Societal Impact:**

The limitation is discussed in Remark 5.2.

**Main Review:**

Originality: This paper studies the perturbed gradient methods for stochastic optimization to escap from saddle points and find local minimum. The new $O(\epsilon^{-3})$ complexity is proved, which is sharper than the revious $O(\epsilon^{-3.5})$ one of SSRGD, and maches the $O(\epsilon^{-3})$ one of SPIDER+DFO$^+$, which is achieved by a more complex negative curvature search. I think this is a novel contribution.

Quality: The theory is technically solid. It is supported by proofs. This is a complete work.

Clarity: This paper is written well. The proof is organized well. The main resutls are proved in only three pages.

Significance: This paper proposes a new technique and achieves the state-of-the-art result by a simpler method. It address a difficult task in a better way than previous work. I think it is a significant job. Other researchers may be likely to follow this work.

I am not familar with the topic of escaping from saddle points in nonconvex stochastic optimization. No comment and criticism for this paper.

**Time Spent Reviewing:**

6 hours

---

> ### Author Response · Authors · 2021-08-10
> **Response to Reviewer Au4C**
>
> Thanks for your positive comments!

---

> > ### Comment · Reviewer_Au4C · 2021-08-25
> > **Reply to the rebuttal**
> >
> > Thanks for the rebuttal.

---

### Official Review · Reviewer_wd5r · 2021-07-18

**Rating:** 5
**Confidence:** 4

**Summary:**

This paper presents a perturbed stochastic gradient framework for solving the stochastic optimization problem and finding local minimum.

**Ethical Concerns:**

I do not have ethical concerns to the work.

**Limitations And Societal Impact:**

I don't see any potential issue related to societal impact. Regarding the limitation of the work, I have already stated it in the main review section.

**Main Review:**

This paper proposed a novel method of perturbed gradient descent and tackled the challenges of finding local minimum for general (nonconvex) optimization problems. The literature review, comparison with existing methods along with the limitations, motivation of the proposed method, and theoretical analysis were very well written and the thought process was very clear. There are a few questions and concerns, and I am listing them here:
1. As authors were trying to answer the question if PGD can get to local minimum with theoretical guarantee, I am wondering what are the most significant contributor to such guarantee. What is the interplay between the Pullback and two phase design?
2. I am not sure if the pullback was fully justified in the theoretical analysis. For example, could you be more specific on the step-size?
3. I was expecting further justification of the proposed method with empirical study.

Overall, I think it would be better to justify pullback with more details in the theory and with empirical study on both the step-size mechanism and the two phase design.


**Time Spent Reviewing:**

6

---

> ### Author Response · Authors · 2021-08-10
> **Response to Reviewer wd5r**
>
> Thanks for your detailed comments!
>
> ----
>
> **Q1:**
> "What are the most significant contributor to such guarantee. What is the interplay between the Pullback and two phase design?"
>
> **A1:**
> As we have stated in Section 4, the most significant contributor to such a guarantee is the use of the 'Pullback' scheme to carefully control the average movement, which leads to a faster convergence guarantee. The two-phase design originates from the original PGD work [13], and the 'Pullback' idea is new and used to control the average movement of two phases, which further controls the approximation error between the estimated gradients and true gradients. Such an interplay has been demonstrated from line 174 to line 183.
>
> ---
>
>
> **Q2:**
> "I am not sure if the pullback was fully justified in the theoretical analysis. For example, could you be more specific on the step-size?"
>
>
> **A2:** The pullback has been justified by the theoretical analysis of Eq.(6.1) from line 221 to 227: the approximation error of the gradient at step $t$ can be bounded by the movement $\tilde O\big(\sum_{i = \lfloor t/q \rfloor q}^{t-1}||x_{i+1} - x_i||^2\big)$, which can be further bounded due to our 'Pullback' scheme: (1) for step $i$ in the **GD phase**, we have $||x_{i+1} - x_i||^2 \leq \eta^2$ because we set $\eta_{t} = \eta/||d_t||$, (2) for step $i$ in **Escape phase**, we have on average, $||x_{i+1} - x_i||^2 \leq \bar{D}$ because we pull the *last* step size $\eta_t$ back to a smaller value, i.e.,  $\eta_t = \sqrt{(t-m_{s})\bar{D} - \sum_{i=m_s+1}^{t-1}\eta_i^2||d_i||^2}/||d_t||$.
>
> ---
>
> **Q3:**  "I was expecting further justification of the proposed method with empirical study."
>
> **A3:** Thanks for your suggestions. We have performed empirical experiments to further justify the proposed algorithm. Please see 'Added Experiments ' at the beginning for more details. From our experiments, we can see that SGD and SPIDER can both get stuck at the saddle point (converge to a point whose objective function value is around 0.6) while PSGD, SSRGD, SPIDER-SFO, Pullback-SPIDER can escape from the saddle point (converge to a point whose objective function value is close to 0).
> Our algorithm Pullback converges the fastest.

---

> > ### Author Response · Authors · 2021-08-30
> > **Followup message**
> >
> > Thank you for your comments and suggestions!  We believe our response has addressed all your comments and questions. Please let us know if you have any other questions, and we are happy to discuss more. If you’re satisfied with our response, we sincerely hope you could reconsider the rating.

---

### Author Response · Authors · 2021-08-10
**Added Experiments**

**Experiment**

We have rigorously proved the convergence guarantee of our algorithm. We originally thought that the empirical results are not needed to verify our results. Thank you for your suggestion. Now we have performed the following experiment to further evaluate our algorithm and compare it with many strong baselines.

We consider the symmetric matrix sensing problem. We need to recover a low-rank matrix $M^*= U^*(U^*)^\top$, where $U^* \in \mathbb{R}^{d\times r}$. We have $n$ sensing matrix $A_i, i \in [n]$ with observation $b_{i} = \langle A_{i}, M^{*}\rangle$. The optimization problem can be written as

$$\min_{U \in \mathbb{R}^{d\times r}}f(U) = \frac{1}{2n}\sum_{i=1}^{n}(\langle A_{i}, UU^{\top}\rangle -b_i)^{2}.$$
For the data generation, we consider $d=50, r=3$ and then generate the unknown low-rank matrix $M^*= U^*(U^*)^\top$ where $U^* \in \mathbb{R}^{d\times r}$ is randomly generated. We then generate $n = 20d$ random sensing matrices  $A_i, i \in [n]$ following standard normal distribution, and thus $b_{i} = \langle A_{i}, M^{*}\rangle$. The global optimal value of the above optimization problem is $0$, because there is no noise in the model.

We evaluate the performance by objective function $||UU^\top - M^*||_F^2/||M^*||_F^2 $ and then report the objective function value versus the number of stochastic gradient evaluations in the following table. We can see that without adding noise or using second-order information, SGD and SARAH/SPIDER are not able to escape from saddle points (i.e., the objective function value of the converged point is far above zero). Our algorithm (Pullback-SPIDER), SSRGD, Perturbed SGD and $\text{SPIDER-SFO}^{+}$(+Neon2) can escape from saddle points. Compared with SSRGD and perturbed SGD, our algorithm converges to the unknown matrix faster.

\# stochastic evaluations | $4e+4$ | $8e+4$ | $1.2e+5$ | $1.6e+5$ | $2e+5$
---- | --- | --- | --- | --- | ---
SGD  | 0.9751 | 0.8444 | 0.7397  | 0.6598 | 0.6192
SPIDER | 0.5886 | 0.6607 | 0.5887 | 0.5963 |  0.6134
PSGD | 0.7290 | 0.3012 | 0.0068 | 0.0057 | 0.0058
$\text{SPIDER-SFO}^{+}$(+Neon2)   | 0.4253 | 0.0218 | 0.0086 | 0.0367 | 0.0690
SSRGD | 0.6306 | 0.2704 | 0.0015 | 3.4907e-04 | 3.5746e-04
Pullback-SPIDER | 0.3668 | 0.0160 | 5.4815e-04 | 3.3121e-04 | 3.3826e-04

---

### Decision · Program_Chairs · 2021-09-27

**Decision:**

Reject

**Comment:**

The paper proposes and analyzes a stochastic first-order optimization method called “Pullback,” proving that it achieves the optimal rate of convergence to $(\epsilon,\sqrt{\epsilon})$-approximate local minima under appropriate assumptions.

After reading the paper and discussing it with the reviewers, I concur with reviewer ADFY’s major concern, that the complexity bound proved in the paper was known under identical assumptions. It is true that prior results on perturbed SGD/AGD also did not strictly innovate on oracle complexity, but I believe the significance of those papers was mostly due to them considering very minor changes to classical algorithms. In contrast, the proposed algorithm seems like a fairly intricate modification of SPIDER and STORM.

Since the proposed theoretical guarantee does not innovate on oracle complexity and does not pertain to a well-known method, its remaining pathway to significance is through showing that the proposed algorithm performs better in practice than existing ones. In their replies, the authors suggest that this is indeed the case, and report some preliminary empirical findings backing these claims. However, the submission itself has no experiments and the information in the authors’ reply is not sufficient for getting a clear picture of the practical performance of the proposed method. For example, the authors do not indicate how the different algorithms are initialized, and whether the poor performance of SGD and SPIDER are related to a pathological initialization. Moreover, a single optimization problem would not make a convincing case for the performance of any optimization method.

With these observations in mind, I encourage the authors to revise and resubmit their paper, placing a much greater emphasis on the practical performance of the algorithm.